# DDLP: Unsupervised Object-centric Video Prediction with Deep Dynamic Latent Particles

**Tal Daniel**  *taldanielm@campus.technion.ac.il*
*Electrical and Computer Engineering*
*Technion - Israel Institute of Technology*

**Aviv Tamar**  *avivt@technion.ac.il*
*Electrical and Computer Engineering*
*Technion - Israel Institute of Technology*

**Reviewed on OpenReview:** *https://openreview.net/forum?id=Wqn8zirthg*

## Abstract

We propose a new object-centric video prediction algorithm based on the deep latent particle (DLP) representation of Daniel & Tamar (2022a). In comparison to existing slot- or patch-based representations, DLPs model the scene using a set of keypoints with learned parameters for properties such as position and size, and are both efficient and interpretable. Our method, *deep dynamic latent particles* (DDLP), yields state-of-the-art object-centric video prediction results on several challenging datasets. The interpretable nature of DDLP allows us to perform "what-if" generation – predict the consequence of changing properties of objects in the initial frames, and DLP's compact structure enables efficient diffusion-based unconditional video generation. Videos, code and pre-trained models are available: `https://taldatech.github.io/ddlp-web/`.

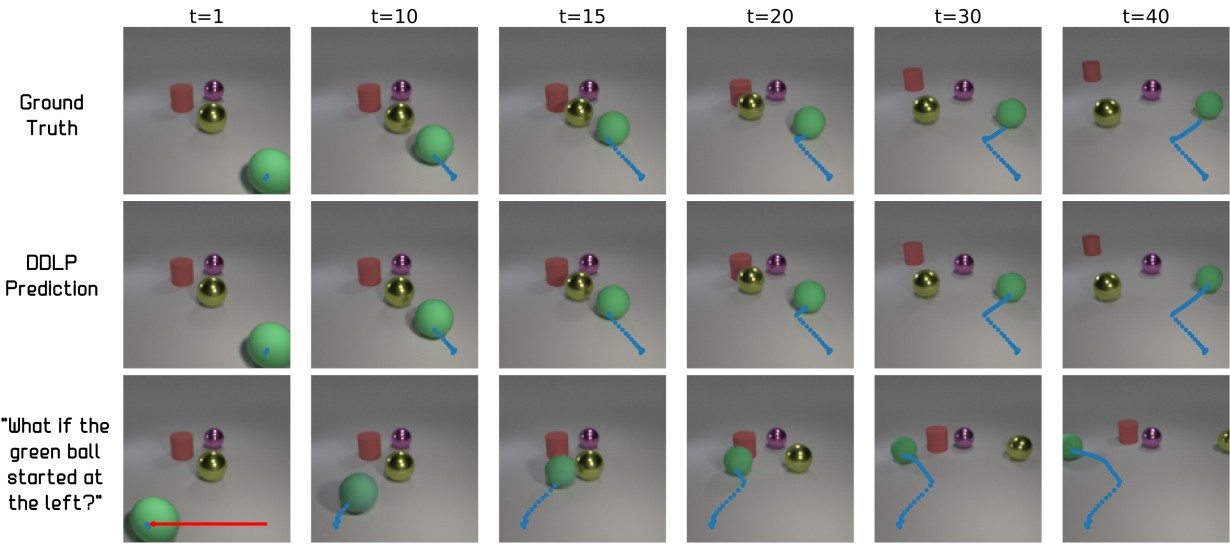

Figure 1: Video prediction on `OBJ3D`. Top: ground-truth video overlaid with DDLP's *inferred* (posterior) trajectory for particle on green ball (blue). Middle: DDLP's generated video conditioned on the first 4 frames, and *predicted* trajectory for particle on green ball (blue). Bottom: a modification in the latent particle space (red arrow) according to the "what if...?" question, and the resulting DDLP video. Note the different trajectory resulting from different collisions due to the modification.

# 1   Introduction

Object-centric models (Burgess et al., 2019; Lin et al., 2020c; Daniel & Tamar, 2022a) leverage a unique inductive bias to represent a scene as a collection of spatially-localized *objects* with physical attributes like size, appearance, and opacity. Applied in diverse contexts such as continuous-state dynamical systems (Battaglia et al., 2016), single images (Lin et al., 2020c; Daniel & Tamar, 2022a), and recently, videos (Wu et al., 2022; Lin et al., 2020b), these models consistently yield more accurate representations compared to general methods modeling pixel space directly. Here, we focus on object-centric video prediction – a challenging domain with potential applications in reinforcement learning (Yoon et al., 2023) and robotics (Zadaianchuk et al., 2022). The object-centric bias entails a tradeoff between expressivity and accuracy. On the one hand, it enforces limitations on the possible videos that can be represented, which for example must be composed of individual objects, with relatively mild scene changes. On the other hand, it has the potential to accurately predict complex physical interactions between several objects, such as the collision depicted in Figure 1 – a difficult task for more general models, even when they are very large (Yan et al., 2021; Yu et al., 2022).

Object-centric video prediction requires learning of two components – object representations and dynamics prediction. Recent work used *off-the-shelf* object representations based on either patches or slots, alongside recurrent neural networks or transformer architectures for dynamics prediction (Lin et al., 2020b; Wu et al., 2022). However, such representations have complexities that limit their performance. Patch-based methods process objects from all patches at training, leading to increased complexity and difficulties in scaling up (Wu et al., 2021b; 2022), while at inference they require a threshold-based filtering process to filter out proposals (Jiang et al., 2019), which requires subtle tuning; slot-based models struggle with real-word datasets (Seitzer et al., 2023) and their time and memory requirements rise rapidly with the number of objects in the scene.

We hypothesise that stronger object representations can significantly advance object-centric video prediction. Our candidate is the recently discovered *deep latent particles* (DLP, Daniel & Tamar (2022a)) representation. DLP employs a set of learned keypoints (particles), identifying areas-of-interest from which object properties are inferred. DLPs therefore strike a balance between a lightweight representation for *all* patches and a heavy representation for a small number of slots, potentially allowing to handle scenes with more objects than slot-based methods, and utilize powerful transformer-based dynamics, which are difficult to scale with patch-based models (Wu et al., 2021b). Further, the DLP representation is interpretable, due to the explicit physical properties of each particle, and enables to control generation by modifying latent space properties of particles.

While Daniel & Tamar (2022a) demonstrated preliminary video prediction with DLPs, by simply extracting particles for each frame individually and learning their dynamics, we find that this approach is not competitive on nontrivial benchmark tasks such as in Figure 1. Our main technical contribution is an extension of DLP to handle dynamic scenes, which we term *deep dynamic latent particles* (DDLP). Intuitively, instead of extracting particles for each frame independently, DDLP learns a *consistent* representation across the whole video. The key challenge, however, is that this requires *tracking* particles in time, and we propose a novel differentiable tracking module so that the representation can be learned end-to-end. To model particle dynamics, we develop a novel *Particle Interaction Transformer* that captures both the spatial and temporal aspects of object interactions. In addition, we propose several technical improvements to DLPs, which further improve generation quality.

In our experiments, we find that DDLP is competitive with or outperforms the state of the art on several challenging object-centric video prediction benchmarks, including scenes with up to 20 objects (difficult for slot-based models), and long videos spanning up to 100 frames (difficult for patch-based models to remain consistent). Notably, DDLP achieves these results while also boasting a lower training memory requirement.

In addition, the finer control that the DDLP representation allows over the image generation enables modifying physical parameters of the initial scene, and answering "what-if" questions based on the generated video, as demonstrated in Figure 1 – the model imagines what would have happened if one of the objects was placed in a different initial position. To the best of our knowledge, this task cannot be solved with existing object-centric models, for which physical properties cannot be extracted and modified easily. Finally, taking advantage of

the compact DDLP representation, we show that DDLP can be combined with diffusion models (Ho et al., 2020) to quickly generate long video scenes unconditionally. To summarize, we propose DDLP, a versatile object-centric video prediction model that is both SOTA on object-centric benchmarks, and enables several new capabilities that may be important for future downstream tasks.

## 2 Related Work

We discuss different unsupervised object-centric latent video prediction studies based on their representation for objects. An extended literature review appears in Appendix C.

Unsupervised object-centric latent video prediction methods largely rely on representations extracted by patch-based methods (Stanić & Schmidhuber, 2019; Crawford & Pineau, 2019; Lin et al., 2020c) or slot-based methods (Burgess et al., 2019; Locatello et al., 2020; Greff et al., 2019; Engelcke et al., 2019; 2021; Kipf et al., 2021; Singh et al., 2022b; Kabra et al., 2021; Singh et al., 2021; 2022a; Anonymous, 2023; Sajjadi et al., 2022; Weis et al., 2021; Veerapaneni et al., 2020) that do not explicitly model objects properties such as position, scale, or depth. STOVE (Kossen et al., 2019), SCALOR (Jiang et al., 2019) and G-SWM (Lin et al., 2020b) belong to the patch-based family of object-centric representations, and use the 'what', 'where', 'depth', and 'presence' latent attributes to represent the objects, and an RNN to model the latent dynamics, where SCALOR and G-SWM add an interaction module to model the interactions between objects. OCVT (Wu et al., 2021b) uses a similar representation, and a Transformer (Vaswani et al., 2017; Radford et al., 2018) is used to model

| Method | Driving Representation | Dynamics Module | Training Scheme |
|---|---|---|---|
| SCALOR (Jiang et al., 2019) | Patch-latent | RNN | End-to-End |
| STOVE (Kossen et al., 2019) | Patch-latent | RNN | End-to-End |
| G-SWM (Lin et al., 2020b) | Patch-latent | RNN | End-to-End |
| OCVT (Wu et al., 2021b) | Patch-latent | Transformer | 2-Stage |
| PARTS (Zoran et al., 2021) | Slot-latent | RNN | End-to-End |
| STEDIE (Nakano et al., 2023) | Slot-latent | RNN | End-to-End |
| SlotFormer (Wu et al., 2022) | Slot-latent | Transformer | 2-Stage |
| DLP (Daniel & Tamar, 2022a) | Particle-latent | GNN | 2-Stage |
| DDLP (Ours) | Particle-latent | Transformer | End-to-End |

Table 1: Unsupervised object-centric video prediction. We categorize models by representation families: patch-latent use all patches of the image to represent objects, slot-latent decompose the image to assigned slots to represent objects and particle-latent use keypoints to represent objects and decompose the image.

the dynamics in a 2-stage training scheme. As the patch-based approaches produce many unordered object proposals, a *matching algorithm* is used to match between objects in consecutive frames before the input to the Transformer, making OCVT hard to scale for real-world datasets. In contrast, our method is trained end-to-end and is based on keypoints, which both significantly reduces the complexity of object decomposition and does not require a matching algorithm to track objects. PARTS (Zoran et al., 2021) and STEDIE (Nakano et al., 2023) use slot-based representations and employ an RNN-based dynamics model of the slots, while OCVP (Villar-Corrales et al., 2023) and SlotFormer (Wu et al., 2022) use a deterministic Transformer to model the dynamics of the slots in a 2-stage training strategy, utilizing the attention mechanism to model the interaction between slots. Differently from SlotFormer, our model uses the more compact latent particles, providing an explicit keypoint-based representation for the objects' location and appearance, which allows for video editing in the latent space, and uses the Transformer as a stochastic prior for latent particles that is trained *jointly* with the particle representation, removing the need to sequentially train two separate models. As we show in our experiments, thanks to the compact representation, our model can fit scenes with many objects and can be utilized for efficient unconditional video generation. Lastly, while the original DLP work (Daniel & Tamar, 2022a) demonstrated results on video prediction, it has limitations. Specifically, particles are extracted independently from each frame, and a simple order-invariant graph neural network is used, which is incapable of modeling complex interactions due to the lack of a tracking module, as discussed in Daniel & Tamar (2022a). Table 1 summarizes the object-centric video prediction approaches.

## 3 Background

**Variational Autoencoders (VAEs):** VAE (Kingma & Welling, 2014) is a learned likelihood-based latent variable model of data $p_\theta(x)$ that maximizes the evidence lower bound (ELBO) using an approximate posterior

distribution $q(z|x)$: $\log p_\theta(x) \geq \mathbb{E}_{q(z|x)} \left[ \log p_\theta(x|z) \right] - KL(q(z|x)\|p(z)) \doteq ELBO(x)$, where the Kullback-Leibler (KL) divergence is $KL(q(z|x)\|p(z)) = \mathbb{E}_{q(z|x)} \left[ \log \frac{q(z|x)}{p(z)} \right]$. The approximate posterior $q_\phi(z|x)$ is also known as the *encoder*, while $p_\theta(x|z)$ is termed the *decoder*. The approximate posterior $q_\phi(z|x)$, likelihood $p_\theta(z|x)$, and prior $p(z)$ are typically modeled as Gaussian distributions. The ELBO can be optimized using the *reparameterization trick*. Following, the term *reconstruction error* and $\log p_\theta(x|z)$ are used interchangeably.

**Deep Latent Particles (DLP):** DLP (Daniel & Tamar, 2022a) is a VAE-based unsupervised object-centric model for images. The key idea in DLP is that the latent space of the VAE is structured as a set of $K$ particles $z = [z_f, z_p] \in \mathbb{R}^{K \times (d+2)}$, where $z_f \in \mathbb{R}^{K \times d}$ is a latent feature vector that encodes the visual appearance of each particle, and $z_p \in \mathbb{R}^{K \times 2}$ encodes the position of each particle as $(x, y)$ coordinates in Euclidean space. Thus, the VAE in DLP is disentangled by construction into a discrete set of interest points in the image. We briefly explain the prior, the encoder, and decoder in DLP. For a complete treatment we refer to Daniel & Tamar (2022a).

The VAE prior $p(z|x)$ in DLP is conditioned on the image $x$, and has a different structure for $z_f$ and $z_p$. For the appearance features $z_f$, a standard zero-mean Gaussian prior is used. For the positions, however, the prior is structured as Gaussians centered on $x-y$ keypoint proposals, with a constant variance. The keypoint proposals are produced by a CNN applied to each patch of the image, followed by a *spatial-softmax* (SSM, Jakab et al. 2018; Finn et al. 2016b) layer to produce a *set* of $x-y$ coordinates, one for each patch. The VAE encoder for $z_p$ is another CNN that maps from the image to means and log-variances of $z_p$. As the posterior keypoints $S_1$ and the prior keypoint proposals $S_2$ are *unordered sets* of Gaussian distributions, the KL term for the position latents is replaced with the Chamfer-KL: $d_{CH-KL}(S_1, S_2) = \sum_{z_p \in S_1} \min_{z_p' \in S_2} KL(z_p\|z_p') + \sum_{z_p' \in S_2} \min_{z_p \in S_1} KL(z_p\|z_p')$. For the appearance features $z_f$, the encoder uses a Spatial Transformer Network (STN, Jaderberg et al. 2015) to apply a small CNN to a region around the position, $z_p$, for every particle. This is termed as a *glimpse*. Finally, for the decoder[1], another glimpse is applied to each particle to create an RGBA image patch around the particle. The patches are combined to create the final image. All components of the DLP model are learned in an unsupervised fashion, by maximizing the ELBO (i.e., minimizing the reconstruction loss and the KL-divergence between the posterior and prior distributions).

## 4 Method

In this section, we give a detailed description of our model, which extends the original DLP (Daniel & Tamar, 2022a) to video, termed *deep dynamic latent particles* (DDLP). We begin with an improvement of the DLP model for single images, which we term DLPv2 (Section 4.1). The DLPv2 model extends the DLP particle representation with more attributes, namely, scale, depth and transparency, and modifies the encoder architecture. We emphasize that DLPv2 can be seen as a standalone replacement for DLP in the single-image setting, as illustrated in Figure 8.

We continue with DDLP – a video prediction model based on DLPv2 (Sections 4.2-4.4). In our generative model, a sequence of frames $x_0, \ldots, x_t$ is generated from a sequence of sets of latent particles $z_0, \ldots, z_t$. A main idea in DDLP is that a Transformer-based dynamics module, which predicts the particles in the next frame, is used only as a *variational prior* during training of the model. The model is trained to reconstruct sequences of $T$ frames, in a VAE fashion, where for reconstructing frame $x_t$, an encoder has access to the frames $x_0, \ldots, x_t$, and outputs a posterior distribution of the particles $z_t$ that compose frame $x_t$, while the prior only has access to frames $x_0, \ldots, x_{t-1}$, and explicitly models the evolution of the particles' distribution $p(z_{t+1}|z_t)$. By the standard VAE loss function, which tries to minimize the KL-divergence between the posterior and prior, the prior is effectively trained to accurately predict the particle evolution. At inference time, we can directly use the Transformer to predict the particles of the next frame, and roll out a complete sequence of frames. To implement this idea successfully, however, we had to solve several technical challenges. The main difficulty is how to align the particles in the posteriors of different frames: treating each frame individually, as in Daniel & Tamar (2022a), does not necessarily yield aligned particles. Our idea is a new *particle tracking posterior*, which uses ideas from the video tracking literature to align the keypoint posteriors in different frames. Other challenges include re-designing the attention matrix in the Transformer to allow

---

[1]Daniel & Tamar (2022a) proposed several decoder architectures. We focus here on their `Object-Based` model.

particles to attend to one another, and finding a suitable positional embedding for the particles in the Transformer. In the following, we detail the components of our method. A high-level specification of our model is illustrated in Figure 2.

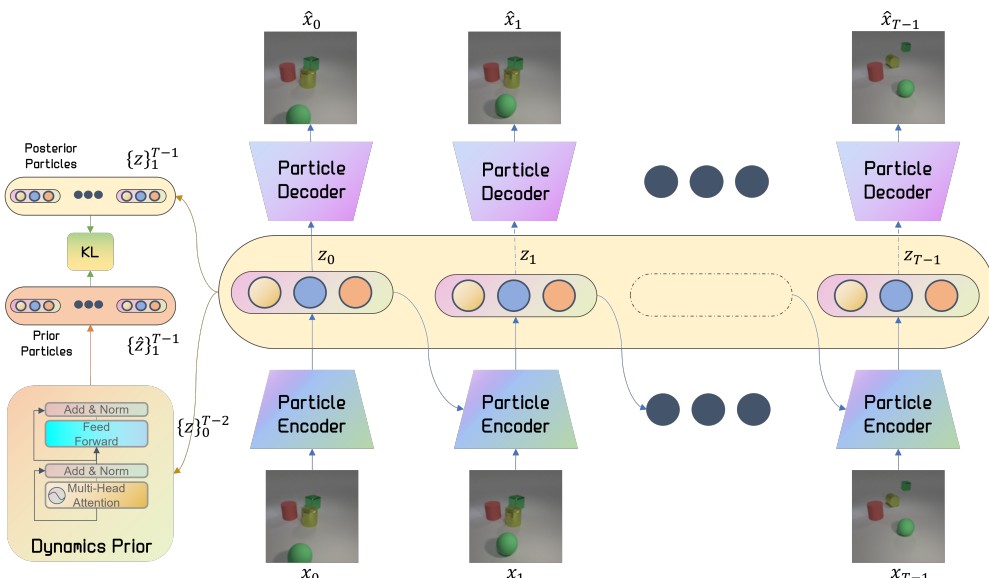

Figure 2: DDLP model illustration. Input are frames $\{x_t\}_{t=0}^{T-1}$. For each $0 \leq t \leq T-1$, a particle encoder produces the latent particle representation $z_t$ given $x_t$ and $z_{t-1}$; this encoder tracks the particles to induce consistency between frames. Each individual latent $z_t$ is fed through the particle decoder to produce the reconstruction of frame $\hat{x}_t$. In addition, a Transformer-based dynamics module models the prior distribution parameters $\{\hat{z}_t\}_{t=1}^{T-1}$ given $\{z_t\}_{t=0}^{T-2}$, and a KL loss term minimizes the distance between the prior $\{\hat{z}_t\}_{t=1}^{T-1}$ and posterior $\{z_t\}_{t=1}^{T-1}$.

## 4.1 DLPv2 - A Modified Latent Particle

We expand upon the original DLP's definition of a latent particle, as described in Section 3, by incorporating additional attributes. This refinement follows recent works in object-centric video modeling (Lin et al., 2020b; Jiang et al., 2019) that demonstrated the importance of explicitly modeling the scale and depth of objects in videos with objects of variable sizes or videos of 3D scenes. In DLPv2, A foreground latent particle $z = [z_p, z_s, z_d, z_t, z_f] \in \mathbb{R}^{6+m}$ is a disentangled latent variable composed of the following learned stochastic latent attributes: position $z_p \sim \mathcal{N}(\mu_p, \sigma_p^2) \in \mathbb{R}^2$, scale $z_s \sim \mathcal{N}(\mu_s, \sigma_s^2) \in \mathbb{R}^2$, depth $z_d \sim \mathcal{N}(\mu_d, \sigma_d^2) \in \mathbb{R}$, transparency $z_t \sim \text{Beta}(a_t, b_t) \in \mathbb{R}$ and visual features $z_f \sim \mathcal{N}(\mu_f, \sigma_f^2) \in \mathbb{R}^m$, where $m$ is the dimension of learned visual features. We illustrate the role of each attribute in Figure 3. Moreover, we assign a single abstract particle for the background that is always located in the center of the image and described only by $m_{\text{bg}}$ latent background visual features, $z_{\text{bg}} \sim \mathcal{N}(\mu_{\text{bg}}, \sigma_{\text{bg}}^2) \in \mathbb{R}^{m_{\text{bg}}}$. Training of DLPv2 is similar to standard DLP, but with modifications of the encoding and decoding that take into account the finer control over inference and generation due to the additional attributes. The main modifications include an additional CNN to infer the latent attributes from particle glimpses, incorporating the scale attribute in the transformation matrix of the STN and factoring the depth attribute when stitching the decoded image. We provide extended details of the modifications in Appendix D.1. The training objective for DLPv2 follows the original DLP. Given an input image $x \in \mathbb{R}^{H \times W \times 3}$, we optimize:

$$\mathcal{L}_{\text{DLP}} = \mathcal{L}_{rec}(x, \tilde{x}) + \beta_{\text{KL}} \left( \text{ChamferKL}(q_\phi(z_p|x) \| p_\gamma(z_p|x)) + \text{KL}(q_\phi(z_{a|f}|x) \| p(z_{a|f})) + \beta_f \text{KL}(q_\phi(z_f|x) \| p(z_f)) \right), \quad (1)$$

where $\tilde{x}$ is the reconstructed image, $\mathcal{L}_{rec}(x, \tilde{x})$ is the reconstruction error, $z_p$ are the posterior keypoints, $p_\gamma(z_p|x)$ denotes the patch-wise spatial-softmax prior of the original DLP, $z_{a|f}$ are all the attributes–scale,

transparency and depth, except for the appearance features $z_f$, and $\beta_{\mathrm{KL}}$ and $\beta_f$ are coefficients to balance the reconstruction loss and the KL-divergence terms.

As the focus of this work is video prediction, we detail the prior, encoder, and decoder of DLPv2 in Appendix D.1. In the following, when referring to individual attributes of a particle in a particular time step, we use a double subscript. E.g., $z_{p,t}$ denotes the position component of the particle at time $t$.

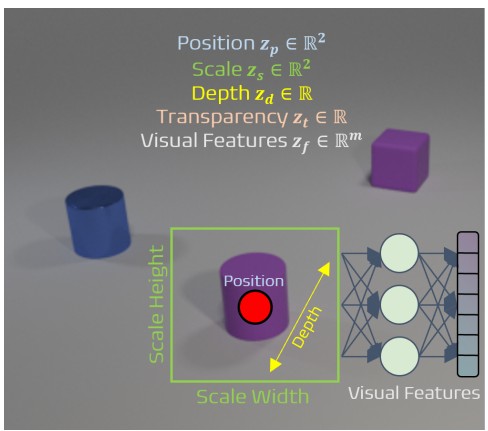

Figure 3: Modified latent particle. DLPv2 introduces extended particle attributes: scale, depth and transparency.

## 4.2 Particle Tracking Posterior

Our goal is to predict the future dynamics of a given video using the latent particle representation. The posterior (encoder) in DLP uses a CNN applied to the image to propose keypoints. However, simply treating each frame individually would not work, as the order of the particles is not aligned between frames and there is no guarantee that objects in consecutive frames will be assigned the same particle, as illustrated in Appendix D.5. Previous attempts to model the dynamics of unordered set of objects resorted to using computationally expensive matching algorithms such as the Hungarian algorithm (Wu et al., 2021b). We propose here a *tracking* approach to achieve particle-alignment between frames, by predicting the keypoints of frame $x_t$ as *offsets* from the keypoints of frame $x_{t-1}$. This idea is inspired by neural tracking algorithms (Held et al., 2016; Harley et al., 2022), which solve a similar problem for object tracking in videos, but here we apply it within an end-to-end training of a video prediction model.

For each particle, our tracking posterior takes as input $p_{t-1}$, the particle position in the previous frame $x_{t-1}$, and searches for $p_t$, the particle position in frame $x_t$ in a patch centered on the previous position (see Figure 4). Since we want to *learn* this tracking behavior, we implement this idea using differentiable components as follows.

Given a sequence of frames $\{x_t\}_{t=0}^{T-1}$ we perform an iterative encoding where in each step (1) the posterior keypoints from the previous step serve as anchors, from which an offset will be learned to the current step as illustrated in Figure 2, and (2) to indicate the region in which the consecutive keypoint may be, the RGB patch input to the encoder is concatenated with an additional single-channel score-map generated by cross-correlating consecutive patches in time as described next.

Our underlying assumption is that the displacement of objects between consecutive frame is small, so the encoder can search for the object in a small region around its previous location (Held et al., 2016). To that end, we simply use $z_{p,t-1}$ as the anchor keypoints for the next frame, i.e., $z_{a,t} = z_{p,t-1}$, where for $t = 0$

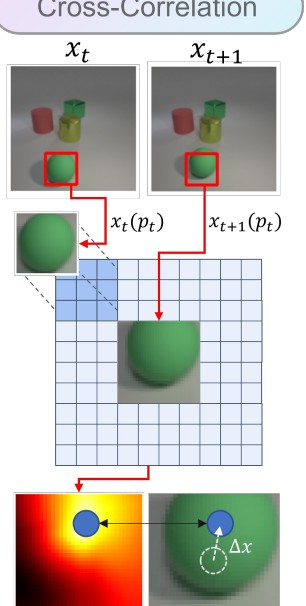

Figure 4: Cross-correlation for tracking. Given a keypoint position $p_t$ for a frame $x_t$, we cross-correlate the patch at $p_t$, $x_t(p_t)$ and the patch from the consecutive frame at the same location $x_{t+1}(p_t)$ to get a heatmap corresponding to the object displacement.

the anchors are keypoint proposals from the prior of a single-image DLP. In addition, to minimize false detections in cases where objects or parts of objects enter the region of the object-of-interest, we compute the normalized cross-correlation (Harley et al., 2022) between the patches, or *glimpses*, from $x_{t-1}$ and $x_t$ around $z_{p,t-1}$, which we denote $x_{t-1}(p_{t-1})$ and $x_t(p_{t-1})$. This is implemented efficiently with group-convolution. This single-channel generated score-map can help direct the search for the object in the consecutive frame as visualized in Figure 4. The score-map is concatenated channel-wise to the input of the encoder component that outputs the particle attributes from each glimpse. We provide more details on the tracking process in Appendix D.5.

## 4.3 Image Decoder

The decoder, closely following the original DLP, reconstructs the scene from the latent particle representation, and is composed of an *object decoder* and a *background decoder*, where the object components and background are stitched together to generate the final reconstructed image. In DDLP, the decoder operates on the latent particles of each frame independently. In Appendix D.4, we detail the decoder's architecture and provide a more technical description of the stitching process.

## 4.4 Generative Predictive Dynamics with Particle Interaction Transformer

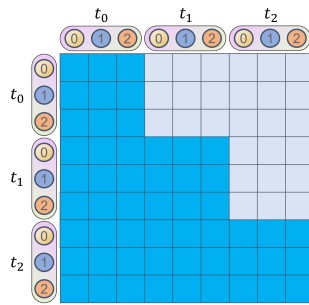

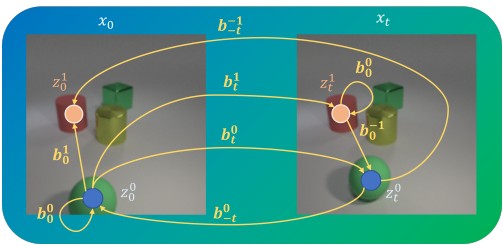

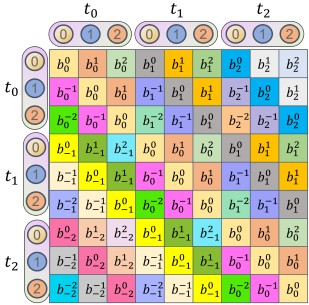

Figure 5: Attention in PINT: particles attend to each other. A causal mask is applied to attend to past and current particles.

Figure 6: Illustration of the relative positional encoding $b_t^i$. Particles in the same time-step $t$ get the same temporal relative embedding $b_t$, while across time, particle $i$ get a unique relative positional embedding w.r.t the other particles $b^i$.

Figure 7: Relative positional bias matrix in PINT. $b_i^j = b_i + b_j$, where $i$ denotes the relative time and $j$ the relative particle position.

The dynamics module models the prior distribution $p_\psi(z_{t+1}|z_t)$ of the particles' temporal evolution. In this work, inspired by the success of Transformers for long-term prediction (Wu et al., 2022; Raffel et al., 2020), we propose a stochastic Transformer-based prior dynamics module that is trained in conjunction with the posterior, and utilizes the attention mechanism to capture the interaction between particles over time in parallel.

Our proposed dynamics module, termed *Particle Interaction Transformer (PINT)*, is a causal Transformer decoder based on the design of Generative Pre-trained Transformer (GPT, (Radford et al., 2018))[2]. Given the set of particles $Z = \{(z_0, ..., z_{K-1}, z_{\text{bg}})_t\}_{t=0}^{T-2}$, where $T$ is the prediction horizon and $z_{\text{bg}}$ is a special particle representing the background (see Appendix D.1), PINT outputs the one-step future prediction $\hat{Z} = \{(\hat{z}_0, ..., \hat{z}_{K-1}, \hat{z}_{\text{bg}})_t\}_{t=1}^{T-1}$. First, the input $Z$ is projected to $G(Z) \in \mathbb{R}^{(K+1) \times T \times D}$, to match the Transformer inner dimension $D$. Differently from previous approaches, we do not add positional embeddings before the attention layers. Instead, building on recent improvements in the literature, we add a learned per-head *relative positional bias* (Shaw et al., 2018; Raffel et al., 2020) which is directly added to the attention matrix before the softmax operation. Empirically, we found this approach more effective than the standard learned positional embeddings that are added prior to the attention layers.

---

[2]We base our implementation on the minGPT open-source implementation (Karpathy, 2021)

However, ignoring the structure of the input and naively adding a positional embedding to each particle may break permutation equivariance among entities, as shown in slot-based approaches (Wu et al., 2022). To that end, we decompose the added relative positional bias to a *temporal* relative positional bias and a *spatial* relative bias. The temporal embedding matrix $B_{\text{time}} \in \mathbb{R}^{T \times T}$ assures that all particles in the same time-step get the same relative temporal encoding, while the spatial embedding matrix $B_{\text{pos}} \in \mathbb{R}^{(K+1) \times (K+1)}$ assures that each particle is assigned a unique relative embedding through time. The relative positional bias is illustrated in Figure 6. $B_{\text{time}}$ and $B_{\text{pos}}$ are repeated $(K+1)$ and $T$ times respectively, and summed to create the relative positional bias matrix $B = B_{\text{time}} + B_{\text{pos}} \in \mathbb{R}^{((K+1)T) \times ((K+1)T)}, B(i,j) = b_i^j = b_i + b^j$, where $i$ denotes the relative time and $j$ the relative particle, as illustrated in Figure 7.

The relative positional bias is added directly to the query-key matrix, i.e., $\text{Attention} = \text{Softmax}\left(\frac{QK^T}{\sqrt{D}} + B\right)V$. To model particle interactions, the attention operation is modified such that particles attend to each other in the same time-step and across time, i.e., the attention matrix $A \in \mathbb{R}^{((K+1)T) \times ((K+1)T)}$, and similarly to GPT we use a causal mask to ensure the attention only considers past inputs, as illustrated in Figure 5. Finally, the attention layers' output is projected with a FC network to the parameters of the prior distribution for the different particle properties. To speed-up training, we use teacher forcing (Williams & Zipser, 1989) at training time, while at inference time, predictions are generated autoregressively. Unless stated otherwise, in all experiments we use a Transformer with 6 layers and 8 heads per layer. A detailed description of PINT's architecture can be found in Appendix D.6.

**Burn-in Frames with the Single Image Prior** To accurately model the dynamics of moving entities and their interactions in a video, a single frame is not enough to extract meaningful dynamics information, such as velocity and acceleration. In DDLP, we define a time-step threshold $\tau$ where for $t \geq \tau$ the output prior parameters by PINT are used to calculate the KL-divergence term between the posterior and prior, while for $t < \tau$, also termed *burn-in frames* (Wu et al., 2022), we use the standard constant prior parameters as in the single-image DLP, detailed in Appendix D.2, to calculate the KL-divergence term (similarly to Wu et al. 2022, we set $\tau = 4$ in all our experiments).

**Training and Implementation Details** Similarly to the training of other VAE-based video prediction models (Denton & Fergus, 2018; Lin et al., 2020b), DDLP is trained to maximize the sum of the temporal ELBO components $\sum_{t=0}^{T-1} ELBO(x_t)$, i.e., minimize the sum of reconstruction errors and KL-divergences over time:

$$\mathcal{L}_{\text{DDLP}} = -\sum_{t=0}^{T-1} \text{ELBO}(x_t) = \sum_{t=0}^{\tau-1} \mathcal{L}_{\text{DLP},t} + \sum_{t=\tau}^{T-1} \mathcal{L}_{\text{dyn},t},$$

where $\mathcal{L}_{\text{DLP},t}$ is defined in Equation 1 and $\mathcal{L}_{\text{dyn},t} = \mathcal{L}_{rec}(x_t, \tilde{x}_t) + \beta_{\text{KL}}\text{KL}(q_\phi(z_t|x)\|p_\psi(z_t|z_{t-1}))$, where here we use the same $\beta_{\text{KL}}$ coefficient as before for *all temporal attributes*. The KL-divergence terms for the distributions we employ have a closed-form formulation, detailed in Appendix E, and for the reconstruction error, we use either the standard pixel-wise MSE or VGG-based perceptual loss (Hoshen et al., 2019), depending on the domain. The reconstruction and KL terms are balanced with a coefficient term $\beta$ (Higgins et al., 2017). DDLP is trained end-to-end, effectively regularizing the posterior particles to be predictable by the learned prior particles, with the Adam (Kingma & Ba, 2014) optimizer and initial learning rate of $2e-4$ which is gradually decreased with a step scheduler. Our method is implemented in PyTorch (Paszke et al., 2017) and we use 1-4 consumer GPUs to train our models. We report the full set of hyper-parameters in Appendix E and provide complexity analysis in Appendix F. Code and pre-trained models are available at: `https://github.com/taldatech/ddlp`.

## 5  Experiments

Our experiments: (1) compare DLPv2 to the original DLP in the single-image setting (Appendix I.1.1);(2) benchmark DDLP on video prediction against the state-of-the-art G-SWM and SlotFormer; (3) demonstrate the capabilities of DDLP in answering *"what if...?"* questions by modifying the initial scenes in the latent space; (4) evaluate our design choices through ablation study; and (5) showcase an application for efficient unconditional video generation by learning a diffusion process in DDLP's latent space.

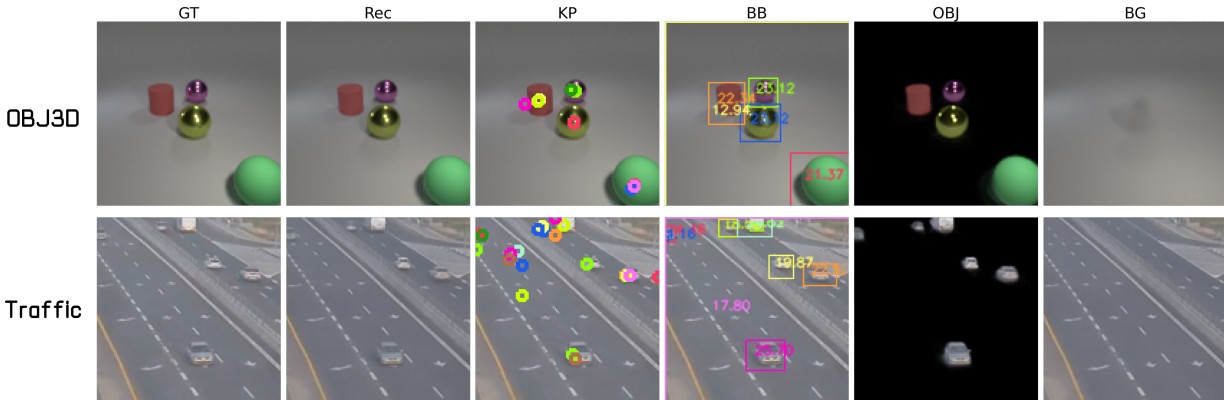

Figure 8: DLPv2 object-centric decomposition. We demonstrate the performance of our proposed improved single-image DLP. Please see Appendix I.1 for more details.

### 5.1 Datasets and Experimental Setup

We evaluate our model on 5 datasets with different dynamics and visual properties[3]. Following is a brief overview of the datasets; an extended description is in Appendix G.

**Balls-Interaction:** (Jiang et al., 2019) A 2D dataset of 3 random-colored balls bouncing and colliding. Videos are 100-frames, $64 \times 64$ pixels. DDLP, similarly to G-SWM, is trained on sequences of 20 frames. At inference, both models are conditioned on 10 frames and predict the next 40. For this dataset, DDLP uses 10 latent particles and the MSE reconstruction loss. As this synthetic dataset includes GT ball positions, we can precisely measure the long-horizon prediction accuracy – a difficult task for video prediction models.

**OBJ3D:** (Lin et al., 2020b) A 3D dataset containing CLEVR-like objects (Johnson et al., 2017). In each 100-frame video of $128 \times 128$ resolution, a random-colored ball is rolled towards random objects of different types positioned in the center of the scene. This dataset has become a standard benchmark for physical understanding as it includes complex object interactions. We train DDLP with 12 particles on sequences of 10 frames with perceptual loss; We train SlotFormer with 6 slots on sequences of 16 frames and G-SWM on sequences of 20 frames as we found it crucial for accurate collision prediction. At inference, all models are conditioned on 6 frames and predict the next 44.

**CLEVRER:** (Yi et al., 2019) A 3D dataset containing CLEVR-like objects. Each video consists of objects entering the scene from various locations, potentially colliding. As new objects may enter the scene at future timesteps, we trim every video to maximize the visibility of objects in the initial scene. We resize the frames to $128 \times 128$ and use sequences of 20 frames for learning, where DDLP is trained with 12 particles and perceptual loss. At inference, we condition on 10 frames and predict the next 90. This dataset is more challenging than **OBJ3D** as there are more variations in the initial conditions.

**PHYRE:** (Bakhtin et al., 2019) A 2D dataset of physical puzzles. We collect simulated data in the form of $128 \times 128$ frames from the BALL-tier tasks in the ball-within-template setting. We train the models on sequences of 20 frames, where DDLP is trained with 25 particles and the MSE reconstruction loss. At inference time, we condition on 10 frames and predict the next 90 frames.

**Traffic:** (Daniel & Tamar, 2022a) Real-world videos of a varying number (up to 20) of cars of different sizes and shapes driving along opposing lanes, captured by a traffic camera. We separate the video to chunks of 50 frames of size $128 \times 128$. For this dataset we train DDLP with 25 particles on sequences of 10 frames with perceptual loss, while SlotFormer is trained with 10 slots on sequences of 16 frames[4] and G-SWM is trained on sequences of 20 frames which was essential to reduce cars from vanishing in the prediction. At inference, all models are conditioned on 6 frames and predict the remaining 44 frames.

---

[3]We did not compare with SlotFormer on **CLEVRER** and **PHYRE** since Wu et al. (2022) reported that special pre-processing is required for SlotFormer to work on these datasets. Further details can be found in Appendix H.

[4]Please see our notes in Appendix H regarding the required compute to train SlotFormer.

## 5.2 Unsupervised Video Prediction Results

We evaluate DDLP on long-term video prediction as described above, while training on short sequences of frames. Our SOTA baselines include the patch-latent based model G-SWM (Lin et al., 2020b), and the slot-based model SlotFormer (Wu et al., 2022), using their publicly available code and pre-trained models (Lin et al., 2020a; Wu et al., 2023).

**Evaluation metrics:** for all datasets, we report the standard visual metrics[5] – PSNR, SSIM (Wang et al., 2004) and LPIPS (Zhang et al., 2018), to quantify the quality of the generated sequence compared to the ground-truth (GT) sequence. In addition, for `Balls-Interaction` we calculate the mean Euclidean error (MED10, Lin et al. 2020b) summed over the first 10 prediction steps as we have GT ball positions.

**Results:** We present our quantitative results in Tables 2 and 3, with additional results and rollouts available in Appendix I. An extended comparison with SlotFormer can be found in Appendix H. Please visit `https://taldatech.github.io/ddlp-web/` to see video rollouts. In the following, we summarize our observations on DDLP's performance in comparison to the baselines.

**Observation 1:** Particle tracking is crucial, and the RPE improves performance, as evidenced by the ablation study (Table 2 bottom). More ablations are reported in Appendix I.8.

**Observation 2:** *DDLP is better at long-term consistency.* In `Balls-Interaction`, where many collisions affect the balls' trajectories, DDLP outperforms G-SWM on all metrics (Table 2). Notably, the accumulated error of the balls' positions is substantially lower, indicating that combining a keypoint representation with a Transformer-based dynamics module yields more accurate predictions.

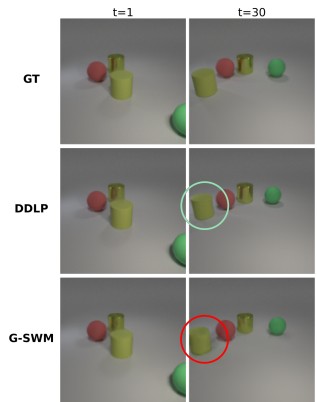

While transformers have been shown to improve patch-based models, their scalability to more complex datasets is limited (Wu et al., 2021b); DLPs are demonstrated to overcome this limitation. Similarly, DDLP outperforms the baselines on all other datasets (with a marginal improvement on `CLEVRER`), as shown in Table 3. We provide example rollouts for the `Traffic` dataset in Figure 10 and for `OBJ3D` in row 2 of Figure 1, demonstrating the superiority of DDLP in long-term prediction. In contrast, objects in SlotFormer's and G-SWM's predictions get blurry, vanish or deform over long horizons. We further demonstrate object deformation in Figure 9.

**Observation 3:** *DDLP performs better in scenes with a large and varying number of objects.* This is particularly evident in `Traffic`, as shown in Figure 10. Note that both DDLP and G-SWM are able to correctly capture the objects, thus, we attribute this to the limited number of slots available in SlotFormer (we used the maximal number of slots available under our compute resources). It is worth noting that DDLP outperforms G-SWM in long-term consistency, as discussed in the previous observation.

Figure 9: Long-term prediction results on `OBJ3D`. While G-SWM tends to deform objects over time (deformed yellow cylinder), DDLP demonstrates better long-term consistency in shape and position.

**Observation 4:** *DDLP produces videos with better perceptual quality* such as more realistic shadows and reflections. Quantitatively, in terms of perceptual quality (LPIPS), DDLP significantly improves upon the baselines ($15 - 35\%$ on `Traffic`, $30\%$ on `PHYRE`, $25\% - 35\%$ on `OBJ3D`, and $8\%$ on `CLEVRER`). This comes with a marginal cost in the distortion measures SSIM and PSNR ($2\%$ on `Traffic`, better performance on `PHYRE`, $1.5\%$ on `OBJ3D`, and $3\%$ on `CLEVRER`)), demonstrating a clear improvement in the perception-distortion tradeoff (Blau & Michaeli, 2018). We attribute these results to the use of the perceptual loss in our model. While it is possible to apply the same loss to G-SWM, for SlotFormer this would require even higher memory requirements. Notably, even when trained with pixel-wise MSE, DDLP outperforms G-SWM in terms of perceptual similarity (LPIPS) for the `Balls-Interaction` and `PHYRE` datasets, as shown in Tables 2 and 3.

**Observation 5:** *DDLP errs when new objects appear during the context window.* Upon analyzing the videos for which DDLP obtains lower LPIPS scores than the baselines, we observed that these are typically scenes where new objects emerge during the context window, which DDLP is not designed to capture. On `CLEVRER`

---

[5]We use the open-source PIQA library(Rozet, 2022).

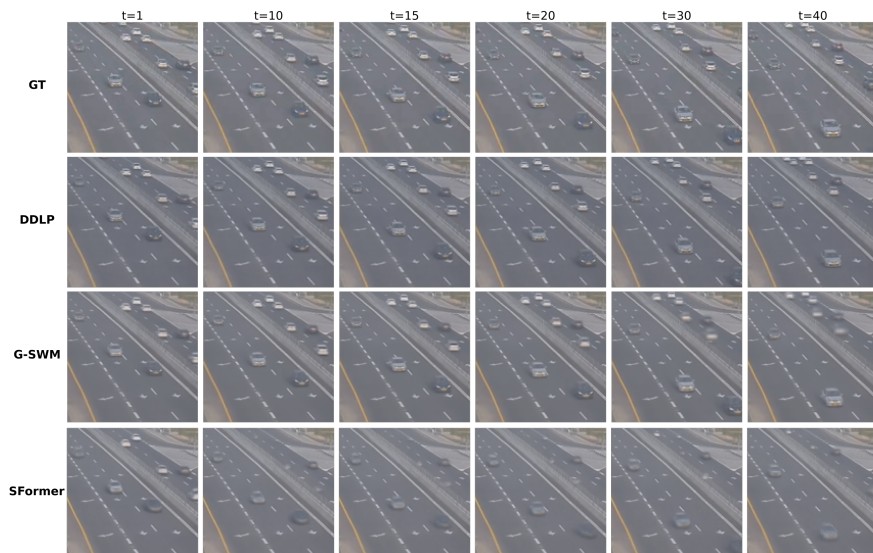

Figure 10: Rollout from the `Traffic` dataset. Models are conditioned on the first 10 frames and predict the next 40. DDLP exhibits sharper predictions compared to G-SWM and SlotFormer. The limited number of slots in SlotFormer results in missed objects. Zoom-in for better observation.

and `Traffic`, for instance, objects can enter the scene at different time-steps, providing G-SWM an advantage because of its object discovery module, as depicted in Figure 12. We further discuss DDLP's limitations in Section 6.

| Balls-Interaction | MED10 ↓ | LPIPS ↓ |
|---|---|---|
| **G-SWM** | 0.256 | 0.23±0.15 |
| **DDLP (Ours)** | **0.149** | **0.18±0.15** |
| −No RPE | 0.157 | 0.194 |
| −DLP (No Tracking) | 2.736 | 0.393 |

Table 2: `Balls-Interaction` dataset comparison. MED10 is the mean euclidean distance error of predicted balls positions, summed over the first 10 steps of generation. Bottom right: Ablation study on the `Balls-Interaction` dataset. In each row, we remove one component from the full model. No RPE – standard positional embedding is used instead of relative positional encoding. DLP (No Tracking) – a standard single-image DLP encoding for all frames instead of tracking.

| Dataset | OBJ3D | | | Traffic | | |
|---|---|---|---|---|---|---|
| | PSNR ↑ | SSIM ↑ | LPIPS ↓ | PSNR ↑ | SSIM ↑ | LPIPS ↓ |
| **G-SWM** | 31.7±6.2 | 0.924±0.05 | 0.118±0.07 | 24.88±1.07 | 0.58±0.07 | 0.235±0.04 |
| **SlotFormer** | 31.2±4.91 | 0.925±0.04 | 0.135±0.05 | 24.93±0.85 | 0.60±0.04 | 0.18±0.03 |
| **DDLP (Ours)** | 31.29±5.22 | 0.923±0.04 | **0.088±0.06** | 24.4±1.02 | 0.61±0.06 | **0.15±0.02** |
| Dataset | CLEVRER | | | PHYRE | | |
| **G-SWM** | 30.12±6.69 | 0.93±0.04 | 0.146±0.07 | 24.64±6.25 | 0.93±0.05 | 0.078±0.06 |
| **DDLP (Ours)** | 29.21±6.16 | 0.92±0.04 | 0.134±0.08 | 26.98±5.3 | 0.95±0.04 | **0.055±0.04** |

Table 3: Quantitative results on video prediction.

## 5.3 *What If...?* Video Prediction

As DDLP provides us with an interpretable latent representation, we can directly modify scenes in the latent space by changing objects' learned attributes – position, scale, depth or shape. Equipped with a pre-trained DDLP on `OBJ3D` from the previous section, we take an initial sequence of frames, encode them to latent particles and manually locate and modify specific particles[6] corresponding to objects-of-interest according to a specific scenario, e.g., "what if the green ball moved to the left?". Then, we unroll the latent sequence with the dynamics module and decode to produce a video of the consequences of our interventions. Row

---

[6]In practice, we apply the same modification (e.g., move a particle by $\Delta x = +0.2$ for the first $\tau$ "burn-in frames" ($\tau = 4$) as we assume small displacements between frames.

3 in Figure 1 demonstrating repositioning of objects, and additional examples in Appendix I.7 controlling scale and visual features, demonstrate that PINT generalizes well to our interventions. We emphasize that "what if" generation exploits DDLP's unique structure, and to our knowledge cannot be obtained with alternative representations such as slot-based models. See `https://taldatech.github.io/ddlp-web/` for video rollouts.

### 5.4 Unconditional Video Generation with DiffuseDDLP

We propose an application of DDLP to unconditional object-centric video generation by learning a diffusion model over the latent particles, termed *DiffuseDDLP*. Our main observation is that training the diffusion model over *particles instead of over pixels* is more efficient, and can exploit the object-centric structure of the problem to produce high quality videos.

We base our model on Denoising Diffusion Probabilistic Model (DDPM, Ho et al. 2020, see Appendix B for an extended description). In DiffuseDDLP we train a DDPM to generate the latent representation of the first $\tau = 4$ frames of a video, $z_\tau \in \mathbb{R}^{\tau \times K \times F}$, where $K$ is the number of particles, and $F$ is the total dimension of the particles' features, using data from a pre-trained DDLP model (cf. Section 5). Then, these generated particles are fed to PINT to predict particles in $T - \tau$ subsequent frames, yielding a sequence of particles $z_T \in \mathbb{R}^{T \times K \times F}$. Finally, the DDLP decoder produces a sequence of images $\hat{x} \in \mathbb{R}^{T \times 3 \times H \times W}$ from the generated sequence of particles.

A key technical finding that we discovered to be important is the structure of the denoising neural network in the diffusion model. Typically, a UNet model is used for DDLP (Ho et al., 2020). However, we find that using DDLP's Transformer-based dynamics module architecture to serve as the DDPM's denoiser yields significantly better results. As our DDPM operates on the compact latent representation of DDLP, training takes less than a day on a consumer-grade GPU, and for inference we are able to generate 100-frame-long videos in a few seconds on a standard CPU.

We train DiffuseDDLP on 3 datasets: `Balls-Interaction`, `OBJ3D` and `Traffic` using the pre-trained DDLPs from Section 5. Figure 11 shows frames from generated videos, demonstrating faithful interactions between the objects and a high visual quality. Please visit `https://taldatech.github.io/ddlp-web/` to view videos. In Figure 14 in the appendix we plot the first $\tau$ frames that were generated by the DDPM diffusion model (decoded to images for visualization purposes), and demonstrate a high temporal consistency between the generated particles. Extended details and more visualizations are available in Appendix B.

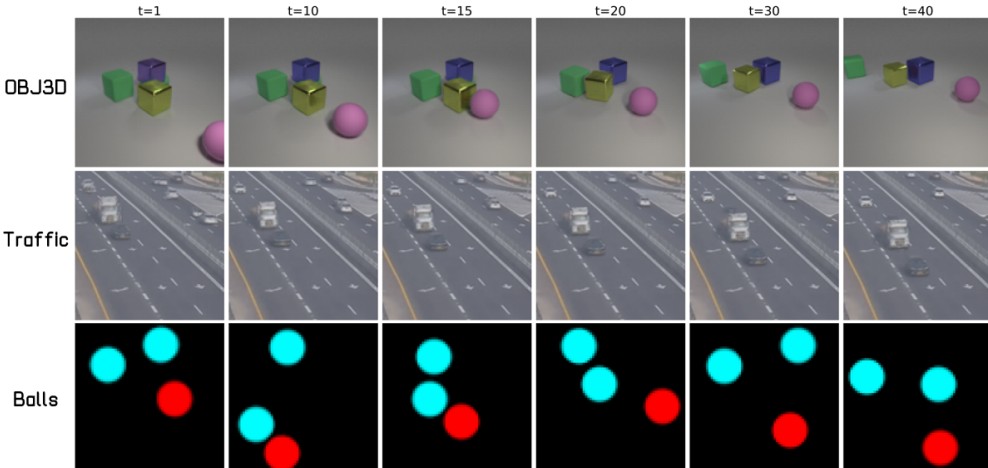

Figure 11: Video generation with DiffuseDDLP. A DDPM diffusion model first generates the first $\tau = 4$ frames represented as latent particles, which are then unrolled to the future with a pretrained PINT dynamics module, and then decoded to pixels with a pretrained DDLP decoder.

# 6   Conclusion and Limitations

DDLP is a new object-centric video prediction model that uses DLPs as an underlying representation, thereby allowing to scale in the dynamics prediction complexity (compared to patches) and number of objects in the scene (compared to slots). We have demonstrated DDLP's SOTA performance in video prediction on various datasets and utilized the explicit latent particles representation to modify videos in the latent space. We see great promise in modeling dynamics with the particles representation as a tool to understand the consequence of interventions, potentially in reinforcement learning where actions are also involved in the physical dynamics.

We conclude with several limitations of DDLP, which may inspire future investigation. Failure cases are further discussed in Appendix J. The assumption that consecutive displacements are small is not valid when new objects enter the scene; we hypothesize that a birth/death process for new particles such as in G-SWM can help mitigate this problem. Our approach does not account for camera movement, i.e., scenes where the background constantly changes and objects appear and disappear between frames. Finally, for videos with objects of various different sizes, the model architecture should be modified to handle multi-scale patches, which we leave for future work.

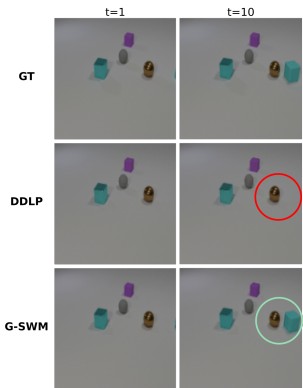

Figure 12: Limitation: DDLP fails to capture new objects that appear during the context window.

## Acknowledgments

This work received funding from the European Union (ERC, Bayes-RL, Project Number 101041250). Views and opinions expressed are however those of the author(s) only and do not necessarily reflect those of the European Union or the European Research Council Executive Agency. Neither the European Union nor the granting authority can be held responsible for them.

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

## Appendix

## Contents

# A    Broader Impact Statement

DDLP is a SOTA object-centric video prediction model that offers an efficient and interpretable solution, outperforming previous methods. The implications of this research may touch upon various societal sectors.

*AI Research*: The proposed model can improve the interpretability of unsupervised scene decomposition and understanding, leading to more transparent AI systems. The public availability of our code and pre-trained models could foster reproducibility, aiding other researchers in building upon this work.

*Autonomous Vehicles and Robotics*: Accurate object-centric video prediction may be important for safe and effective operation of robots. DDLP might aid in faster, interpretable decision-making and risk assessment.

*Healthcare*: DDLP may also find applications in medicine, for example in analyzing medical videos which can be decomposed to interacting objects, assisting in downstream tasks of diagnosis and treatment.

Like other generative algorithms, this model could be exploited for misuse. The interpretability of DDLP could help in this regard, allowing for detection and regulation mechanisms to be developed.

# B    DiffuseDDLP: Unconditional Video Generation with Latent Particle Denoising Diffusion

In this section, we propose to extend DDLP to unconditional object-centric video generation by learning a diffusion model over the latent particles. We base our model on Denoising Diffusion Probabilistic Model (DDPM, Ho et al. (2020)) and the publicly available code base (Wang, 2022). Our DDPM employs a pre-trained DDLP model to generate the latent representation of the first few frames of a video, which are then used as input to the conditional dynamics module of DDLP that unrolls the latent representation of the video into the future, and decodes them back to pixel-space with DDLP's decoder. Crucially to the success of our approach is the design of the denoiser–we adapt DDLP's Transformer-based dynamics module architecture to serve as the DDPM's denoiser instead of the typical UNet for 1D and 2D data. As our DDPM operates on the compact latent representation of DDLP, training converges in less than a day and we are able to generate 100-frame-long videos in a few seconds on a standard CPU.

## B.1    Background: Denoising Diffusion Probabilistic Models (DDPMs)

Denoising Diffusion Probabilistic Models (DDPMs) are probabilistic models that aim to capture the density of a given dataset by assuming that it arises from a diffusion process. The diffusion process is modeled as a sequence of invertible diffusion steps, with each step being parameterized by a time-dependent noise level.

**Forward Diffusion Process:** The forward diffusion process begins with a data point sampled from a real data distribution, $x_0 \sim q(x)$. In this process, a Gaussian noise, with a variance schedule $\beta_t \in (0,1)_{t=1}^T$, is added to the sample in $T$ steps. This produces a sequence of noisy samples, $x_1, ..., x_T$. Mathematically, the forward diffusion process is represented as:

$$q(x_{1:T}|x_0) = \prod_{t=1}^{T} q(x_t|x_{t-1}),$$

$$q(x_t|x_{t-1}) = \mathcal{N}(x_t; \sqrt{1-\beta_t}x_{t-1}, \beta_t I).$$

As $\beta_t \to 1$, $x_T$ is equivalent to an isotropic Gaussian distribution. The forward process allows $x_t$ to be sampled at any arbitrary time step $t$ using the closed-form solution: $q(x_t|x_0) = \mathcal{N}(x_t; \sqrt{\bar{\alpha}_t}x_0, (1-\bar{\alpha}_t)I)$, where $\bar{\alpha}_t = \prod_{i=1}^{t}(1-\beta_i)$.

**Reverse Diffusion Process:** Given a sample from a Gaussian noise, $x_T \sim \mathcal{N}(0, I)$, the goal of the reverse diffusion process is to model $q(x_{t-1}|x_t)$ to reach a true sample $x_0$. However, this is intractable, and therefore, the reverse process in DDPM approximates these conditional probabilities with a learned model, $p_\theta$, which is defined as:

$$p_\theta = p(x_T) \prod_{t=1}^{T} p_\theta(x_{t-1}|x_t),$$

$$p_\theta(x_{t-1}|x_t) = \mathcal{N}(x_{t-1}; \mu_\theta(x_t, t), \Sigma_\theta(x_t, t)).$$

**Loss Function:** During training, the Gaussian noise term is reparameterized, and a neural network is trained to predict the added noise $\epsilon_t$ from the input $x_t$ at time step $t$. The loss term $L_t$ minimizes the difference from $\tilde{\mu}_t = \frac{1}{\sqrt{\alpha_t}}\left(x_t - \frac{1-\alpha_t}{\sqrt{1-\bar{\alpha}_t}}\epsilon_t\right)$. Mathematically, the loss function is represented as:

$$L_t = \mathbb{E}_{x_0, \epsilon}\left[A||\epsilon_t - \epsilon_\theta(\sqrt{\bar{\alpha}_t}x_0 + \sqrt{1 - \bar{\alpha}_t}\epsilon_t, t)||_2^2\right],$$

where $A = \frac{(1-\alpha_t)^2}{2\alpha_t(1-\bar{\alpha}_t)||\Sigma_\theta||_2^2}$. However, in practice, it was found that DDPMs work better without the $A$ term, simplifying the loss function and removing the need to model $\Sigma_\theta$.

## B.2 DiffuseDDLP

We propose a diffusion-based approach for unconditional object-centric video generation termed *DiffuseDDLP*. Our approach employs a DDPM over the latent space of a pre-trained DDLP as follows: given a sequence of $\tau$ input frames $x_0, ..., x_{\tau-1}$, we encode a sequence of latent particles $z_\tau = \{z_K^t\}_{t=0}^\tau \in \mathbb{R}^{\tau \times K \times F}$ with DDLP's frozen encoder, where $K$ is the number of particles, including a background particle, and $F$ is the total dimension of the concatenated attributes (position, scale, depth, transparency and visual features). Then, we train a DDPM model to approximate the distribution of the latent particles sequence of length $\tau$. Equipped with a trained DDPM, we sample sequences of $\tau$ particles and use them as input to the pre-trained DDLP dynamics module, PINT, to unroll the sequence to horizon $T$, where $T >> \tau$ (we use $\tau = 4$ for all datasets). Finally, we generate a video by decoding the sequence of generated particles back to pixel-space with DDLP's pre-trained image decoder. The overall process is depicted in Figure 13.

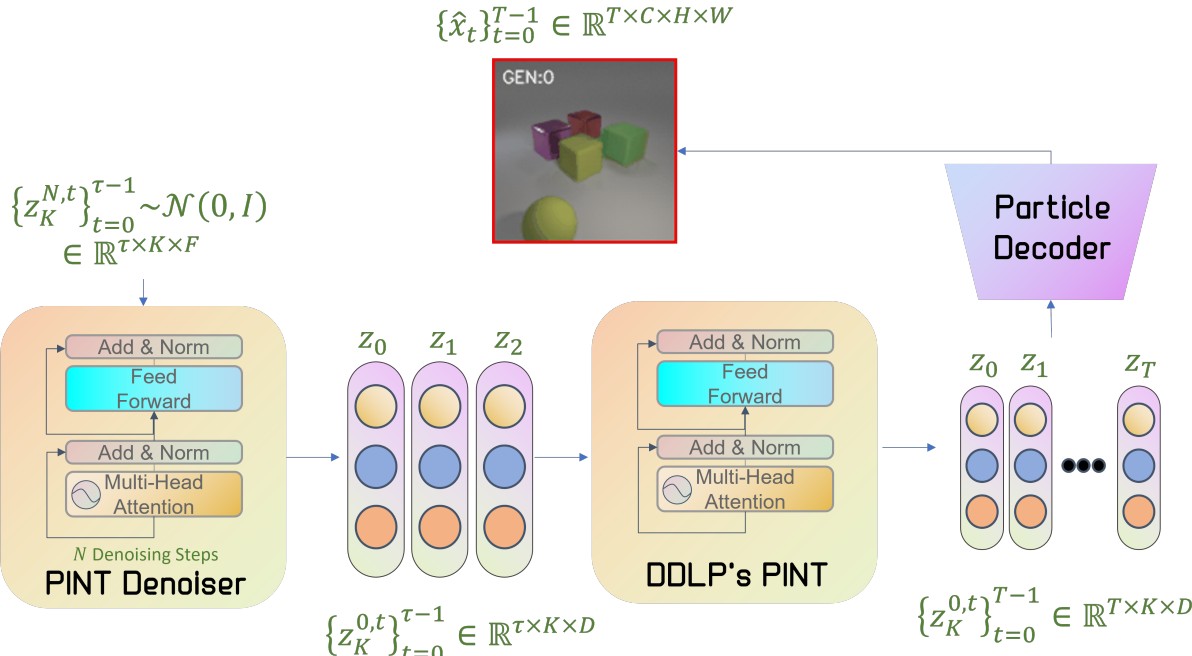

Figure 13: DiffuseDDLP model illustration. We learn a DDPM to generate the first $\tau$ frames represented as latent particles, $z_\tau \in \mathbb{R}^{\tau \times K \times F}$, where $K$ is the number of particles, and $F$ is the total dimension of the particle's features. Then, we unroll the particles evolution over $T$ steps using DDLP's PINT, $z_T \in \mathbb{R}^{T \times K \times F}$, and decode them to back to pixel-space with DDLP's image decoder to generate a video $\hat{x} \in \mathbb{R}^{T \times 3 \times H \times W}$.

**Denoiser Architecture:** The typical denoiser architecture is based on a 1D or 2D UNet. However, due to the unique structure of DDLP's latent space, a denoiser architecture that considers the evolution of a

set of particles over time is required. In our initial experiments, both the 1D and 2D UNet introduced unsatisfying visual artifacts, with the 1D UNet performing slightly better. However, we find that, the same transformer-based architecture that was introduced in DDLP, PINT, performs well in predicting the dynamics of the particles. In DiffuseDDLP, we repurpose PINT to perform denoising instead of predicting the next state of the particles. We keep the same architecture and positional bias that were introduced in DDLP, ensuring that the denoiser is optimized for DDLP's unique structure.

### B.3 Unconditional Video Generation with DiffuseDDLP

We train DiffuseDDLP on 3 datasets: `Balls-Interaction`, `OBJ3D` and `Traffic` using the pre-trained DDLPs from Section 5. In all our experiments, we generate the first $\tau = 4$ frames represented as latent particles, $z_\tau \in \mathbb{R}^{\tau \times K \times F}$, with the learned DDPM, unroll their evolution over $T = 100$ steps using DDLP's PINT, $z_T \in \mathbb{R}^{T \times K \times F}$, and decode them to back to pixel-space with DDLP's image decoder to generate a video $\hat{x} \in \mathbb{R}^{T \times 3 \times H \times W}$. We trained DDPM for $100,000$ iterations and used $N = 1000$ denoising steps in DDPM both during training and inference. As DDLP's latent space is compact, training a DDPM is fast and takes less than a day on a single RTX A4000 GPU. For example, for `OBJ3D`, each frame is represented as 12 foreground particles and 1 background particle, i.e. $K = 13$, and each particle has a total of $F = 14$, resulting in $\tau \times K \times F = 4 \times 13 \times 14 = 728$, which is much smaller than a single CIFAR10 image, $C \times H \times W = 3 \times 32 \times 32 = 3072$.

**Results:** The results in Figure 14 demonstrate that the DDPM in DiffuseDDLP generates temporally consistent latent particles (decoded to images for visualization purposes), which are then unrolled to the future with DDLP's PINT, resulting in faithful interaction dynamics as depicted in Figure 11. Moreover, the generated videos showcase the high quality of the generated content. Please visit `https://taldatech. github.io/ddlp-web/` to view the video generations.

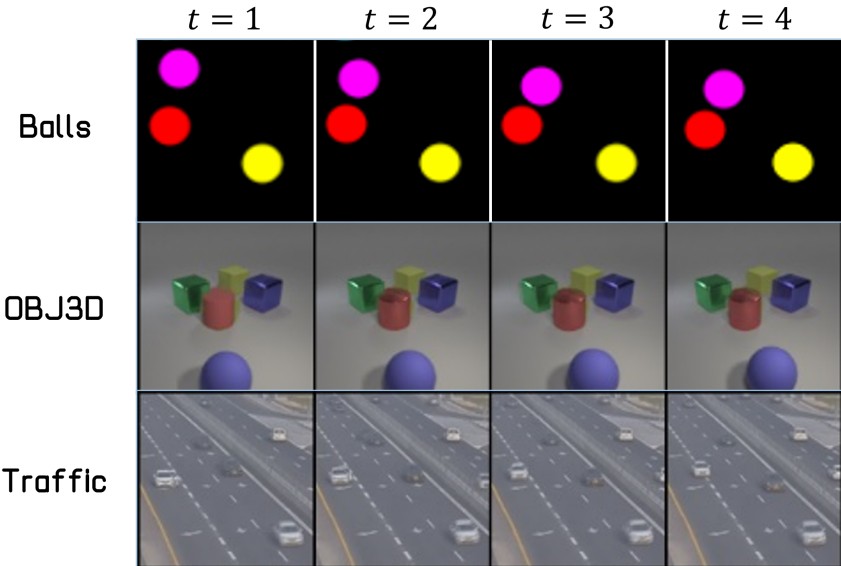

Figure 14: Generation results of the first $\tau = 4$ frames represented as latent particles by the DDPM of DiffuseDDLP, decoded to pixel-space with DDLP's image decoder for visualization purposes.

## C Extended Related Work

We discuss different video prediction studies based on their representation for objects: keypoints, object-centric models, and general latent vector-based representations.

**General unsupervised latent video prediction:**[7] traditionally, latent-based video prediction is composed of encoding visual observations into a compact latent space, predicting the next latents using a recurrent dynamics module, and decoding them back into images. Early models (Finn et al., 2016a; Ebert et al., 2017; Villegas et al., 2017; Lee et al., 2018; Denton & Fergus, 2018) learn latent representations with a convolutional-based architecture and an RNN-based dynamics module to model the latent dynamics, while recent attempts to improve long-horizon prediction use discrete latent space (Walker et al., 2021) or focus scaling-up via hierarchical modeling or larger architectures (Villegas et al., 2019; Wu et al., 2021a; Wang et al., 2022). All aforementioned methods model the dynamics based on a representation extracted by considering the whole scene at once, without explicit decomposition to objects, which leads to blurriness and objects disappearing in the long-term prediction (Wu et al., 2022). Several recent works (Nash et al., 2022; Yu et al., 2022; Yan et al., 2021) model the dynamics using self attention (Vaswani et al., 2017; Dosovitskiy et al., 2020). While these methods improved the visual quality of the generated videos, they do not explicitly model objects and are outperformed by object-centric models on data involving complex physical interactions (Wu et al., 2022).

**Keypoint-based unsupervised video prediction:** the idea of utilizing keypoint-based latent representations has been explored in several works. Kim et al. (2019) learn a class-guided video prediction model by learning a KeyNet (Jakab et al., 2018)-based keypoint detector and then training a motion generator implemented as a recurrent adversarial conditional VAE. Similarly, (Minderer et al., 2019) and (Gao et al., 2021) use KeyNet to learn keypoints and propose a variational RNN as a prior to model stochastic dynamics, where the latter further maps the keypoints onto a discrete binary grid for improved performance. While in essence we strive to achieve a similar goal, all previous KP-based approaches do not explicitly model object properties and interactions and the visual features are not represented as random latent variables, but derived from the source frame sequence feature maps. This naturally leads to blurriness or disappearing objects in long-term predictions (Daniel & Tamar, 2022a). On the other hand, V-CDN (Li et al., 2020) first detect unsupervised keypoints with Transporter (Kulkarni et al., 2019), from which a causal graph is built and then performs video prediction of physical interaction with an Interaction Network (Battaglia et al., 2016). Lastly, the original DLP (Daniel & Tamar, 2022a) has shown promising results for video prediction from latent particles on a real-world dataset; however, a simple graph neural network is used that is incapable of modeling complex interactions.

**Unsupervised object-centric latent video prediction:** these methods largely rely on object-centric representations extracted by patch-based methods (Stanić & Schmidhuber, 2019; Crawford & Pineau, 2019; Lin et al., 2020c) or slot-based methods (Burgess et al., 2019; Locatello et al., 2020; Greff et al., 2019; Engelcke et al., 2019; 2021; Kipf et al., 2021; Singh et al., 2022b; Kabra et al., 2021; Singh et al., 2021; 2022a; Anonymous, 2023; Sajjadi et al., 2022; Weis et al., 2021; Veerapaneni et al., 2020) STOVE (Kossen et al., 2019), SCALOR (Jiang et al., 2019) and G-SWM (Lin et al., 2020b) belong to the patch-based family of object-centric representations, and use the 'what', 'where', 'depth', and 'presence' latent attributes to represent the objects, and an RNN to model the latent dynamics, where SCALOR and G-SWM add an interaction module to model the interactions between objects. OCVT (Wu et al., 2021b) uses a similar representation, and a Transformer (Vaswani et al., 2017; Radford et al., 2018) is used to model the dynamics in a 2-stage training scheme. As the patch-based approaches produce many unordered object proposals, a matching algorithm is used to match between objects in consecutive frames before the input to the Transformer, making OCVT hard to scale for real-world datasets. In contrast, our method is trained end-to-end and is based on keypoints, which both significantly reduces the complexity of object decomposition and does not require a matching algorithm to track objects. PARTS (Zoran et al., 2021) and STEDIE (Nakano et al., 2023) use slot-based representations and employ an RNN-based dynamics model of the slots, while OCVP (Villar-Corrales et al., 2023) and SlotFormer (Wu et al., 2022) use a deterministic Transformer to model the dynamics of the slots in a 2-stage training strategy, utilizing the attention mechanism to model the interaction between slots. Differently from SlotFormer, our model uses the more compact latent particles, providing an explicit keypoint-based representation for the objects' location and appearance, which allows for video editing in the latent space, and uses the Transformer as a stochastic prior for latent particles that is trained *jointly* with the particles representation, removing the need to sequentially train two separate models.

---

[7]We use the term "latent video prediction" to refer to models where the video prediction is based on a latent encoding of the original frames.

As we show in our experiments, thanks to the compact representation, our model can fit scenes with many objects. Table 1 summarizes the object-centric video prediction approaches.

# D Method - Extended Details

In this section, we go into the finer details of our model with accompanying visualizations and illustrations. We begin with our modified definition of a *latent particle*, followed by an overview of the structured keypoints prior used in DLP and how to encode latent particles from a single image. Next, we extend the encoding scheme to a video, i.e., a sequence of image frames, to allow tracking particles over time. Then, we detail how to reconstruct (decode) images from particles. Finally, we construct a Transformer-based dynamics module that serves as a stochastic dynamics prior for the particles' evolution over time.

## D.1 DLPv2 - A Modified Latent Particle

For a more accurate prediction and to improve the autoencoding performance, we extend the definition of a latent particle from the original DLP as described in Section 3 with additional attributes, following recent works in object-centric video modeling (Lin et al., 2020b; Jiang et al., 2019). A foreground latent particle $z = [z_p, z_s, z_d, z_t, z_f] \in \mathbb{R}^{6+m}$ is a disentangled latent variable composed of the following learned stochastic latent attributes: position $z_p \sim \mathcal{N}(\mu_p, \sigma_p^2) \in \mathbb{R}^2$, scale $z_s \sim \mathcal{N}(\mu_s, \sigma_s^2) \in \mathbb{R}^2$, depth $z_d \sim \mathcal{N}(\mu_d, \sigma_d^2) \in \mathbb{R}$, transparency $z_t \sim \text{Beta}(a_t, b_t) \in \mathbb{R}$ and visual features $z_f \sim \mathcal{N}(\mu_f, \sigma_f^2) \in \mathbb{R}^m$, where $m$ is the dimension of learned visual features. Moreover, we assign a single abstract particle for the background that is always located in the center of the image and described only by $m_{\text{bg}}$ latent background visual features, $z_{\text{bg}} \sim \mathcal{N}(\mu_{\text{bg}}, \sigma_{\text{bg}}^2) \in \mathbb{R}^{m_{\text{bg}}}$.

The motivation for the additional attributes follows previous works (Lin et al., 2020b; Jiang et al., 2019) that demonstrated the importance of explicitly modeling the scale and depth of objects in videos with objects of variable sizes or videos of 3D scenes. Next, we motivate the role of each attribute as illustrated in Figure 3.

**Position** $z_p \in \mathbb{R}^2$**:** describes the spatial location of the particle, i.e., its $x - y$ coordinates in $[-1, 1]$. Similarly to other object-centric models (Lin et al., 2020b; Jiang et al., 2019), $z_p$ is a modeled as a Gaussian distribution $\mathcal{N}(\mu_p, \sigma_p^2)$. As in DLP, the prior for the position is based on SSM from patches, which constrains $z_p$ to have a spatial meaning.

**Scale** $z_s \in \mathbb{R}^2$**:** describes the height and width of the bounding box around the particle. Similarly to other object-centric models, $z_s$ is modeled as a Gaussian distribution $\mathcal{N}(\mu_s, \sigma_s^2)$, and is activated with a Sigmoid function to constrain the values to be in $[0, 1]$.

**Depth** $z_d \in \mathbb{R}$**:** describes the depth of the particle. Its main role is to define the order in which the decoded objects are stitched together when reconstructing the image. Similarly to G-SWM (Lin et al., 2020b), $z_d$ is modeled as a Gaussian distribution $\mathcal{N}(\mu_d, \sigma_d^2)$.

**Transparency** $z_t \in \mathbb{R}$**:** describes the degree to which a particle's features are part of the scene, a value in $[0, 1]$. When $z_t = 0$, the object is completely transparent and not part of the scene, when $z_t = 1$, the object is completely visible, and when $z_t \in (0, 1)$ the object is partially visible. Unlike previous works which model $z_t$ as a "presence" variable sampled from a Bernoulli distribution, in DLPv2 we model $z_t$ as a Beta distribution, namely $\text{Beta}(a_t, b_t)$. There are two main benefits of doing so: (1) unlike Bernoulli distribution which is discrete and requires relaxations or gradient estimations to be differentiable, Beta distribution is continuous and supports differentiable sampling via reparameterization, making the optimization process simpler, and (2) allowing continuous values for the transparency in $(0, 1)$ can naturally model partially visible objects in the scene. Fortunately, similarly to Gaussian and Bernoulli distributions, Beta distribution has a closed-form KL-divergence solution.

**Features** $z_f \in \mathbb{R}^m, z_{\text{bg}} \in \mathbb{R}^{m_{\text{bg}}}$**:** describes the particle's visual appearance, i.e, features of its surrounding region. Similarly to other object-centric models, $z_f$ and $z_{\text{bg}}$ are modeled as Gaussian distributions $\mathcal{N}(\mu_f, \sigma_f^2)$ and $\mathcal{N}(\mu_{\text{bg}}, \sigma_{\text{bg}}^2)$, respectively.

### D.2 DLPv2 - Patch-wise Conditional Prior

Similarly to the original DLP, to disentangle position from appearance of the particles, we consider a conditional VAE formulation (Sohn et al., 2015), where a prior for the keypoints is explicitly learned given an image $x$, $p_\gamma(z|x) = p_\gamma(z_p|x) \times p_\gamma(z_a|x)$, where $z_a = [z_s, z_d, z_t, z_f]$ denotes the latent variable for all non-position particle attributes. The prior distribution parameters, means and covariances for Gaussian distributions, and $a, b$ in the Beta distribution, for the rest of the attributes $z_a$ are not learned, i.e., they are hyper-parameters.

To generate a set of prior keypoints, termed *keypoint proposals* we follow a patch-based approach, which was found to work well in prior work (Jiang et al., 2019; Lin et al., 2020b; Daniel & Tamar, 2022a). The input $x \in \mathbb{R}^{H \times W \times 3}$ is split into $K_p$ patches of size $D \times D$ (usually $D \in \{8, 16\}$), and for each patch, a small convolutional neural network (CNN) followed by a spatial-softmax (SSM) layer outputs a single keypoint proposal. A similar process of unsupervised keypoint detection via SSM has been proposed in Jakab et al. (2018); however, in DLP there are two main differences: (1) the process is applied over patches and not the whole image, producing a single keypoint per-patch using a single heatmap as input to SSM and (2) the output only serves as prior and not directly used to reconstruct the image.

In practice, as the set of proposals can grow large with the number of patches, we *learn* to filter the keypoints to produce a subset of $L$ prior keypoints, where $L$ is a hyper-parameter. Differently from the original DLP, where the $L$ chosen keypoints were the top-$L$ distant keypoints from the center of their respective center, in DLPv2 the top-$L$ keypoints with the smallest sum of empirically-calculated SSM variance $s = \sigma_x^2 + \sigma_y^2 + \sigma_{xy}$ (Sun et al., 2022), are chosen. The motivation for using the variance is that the SSM operation detects areas-of-interest based on the activation heatmap, and when an object, or part of it, is found, the activation is tightly centered around it, i.e., sharp softmax values, as described next. Empirically, using the SSM variance resulted in better detection of small objects than the original distance heuristic of Daniel & Tamar (2022a). The architecture of the proposed prior is illustrated in Figure 16.

The process is formally described as follows. Given a heatmap $H \in \mathbb{R}^{H \times W}$, which is the output of a CNN applied to an image patch, the softmax function is applied spatially over the $H$ and $W$ dimensions to produce probabilities for each coordinate. Using these values, we calculate the expected coordinate values for each axis, and their covariance following (Sun et al., 2022). Given a softmax-activated normalized heatmap $H \in \mathbb{R}^{H \times W}$, where each value $h_{ij} = H(i, j)$ indicates the probability of coordinate $(i, j)$ to be a keypoint, we are interested in extracting the expected coordinate, the mean, $(\mu_x, \mu_y)$ and the coordinate covariance values $\sigma_x^2, \sigma_y^2$ and $\sigma_{xy}$ as follows:

$$\mu_x = \sum_i x_i \sum_j H(i, j)$$

$$\mu_y = \sum_j y_j \sum_i H(i, j)$$

$$\sigma_x^2 = \sum_{ij} (x_i - \mu_x)^2 H(i, j)$$

$$\sigma_y^2 = \sum_{ij} (y_j - \mu_y)^2 H(i, j)$$

$$\sigma_{xy} = \sum_{ij} (x_i - \mu_x)(y_j - \mu_y) H(i, j)$$

Note that $\sum_j H(i, j)$ and $\sum_i H(i, j)$ are the marginalized distributions per-axis. The SSM operation is illustrated in Figure 15. Intuitively, in regions where the activation is sharp, we expect the softmax to produce a distribution with a small covariance values, which may indicate a non-background area-of-interest, e.g., objects, lines or corners, making the low-covariance a suitable criterion for detection and filtering. Note that the SSM operation is differentiable, i.e., the heatmap values are learned according to reconstruction

objective of the images, thus leading the SSM operation to locate regions of the image that are crucial for an appropriate reconstruction of the scene.

**Architecture details:** the architecture of the prior encoder, following the original DLP, is a small CNN that is fed with the patches, structured has a 3 layers, ReLU activated with 4-group Group Normalization, with $[16, 32, 64]$ or $[32, 32, 64]$ channels in each layer, ending with a $1 \times 1$ convolution mapping to a single channel output per patch, where SSM is applied over the 2D feature map output.

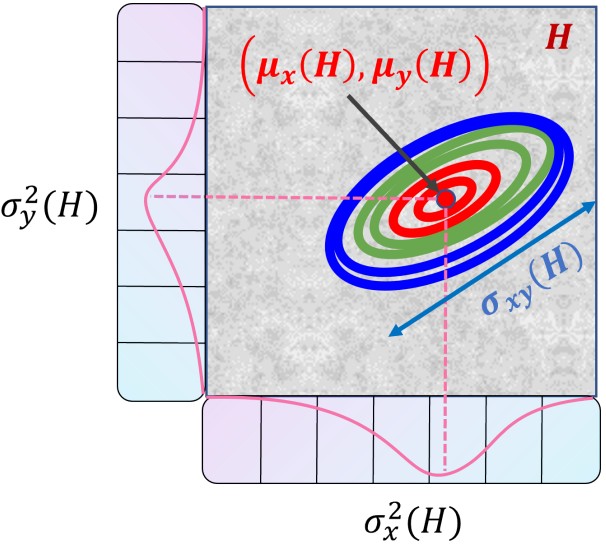

Figure 15: Spatial-softmax. Given a heatmap $H \in \mathbb{R}^{H \times W}$, which is the output of a CNN, the softmax function is applied spatially over the $H$ and $W$ dimensions to produce probabilities for each coordinate. Using these values, we calculate the expected coordinate values for each axis, and their covariance.

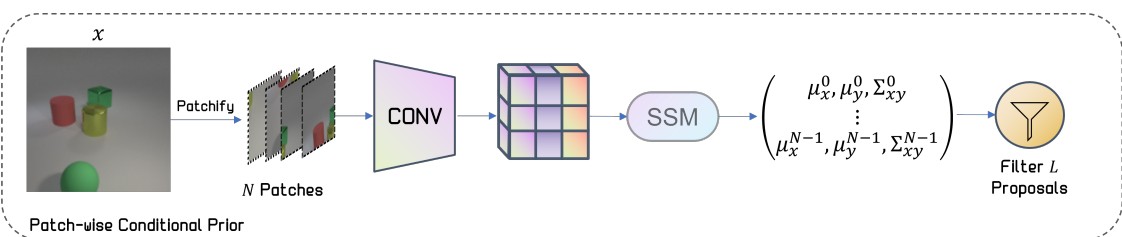

Figure 16: Patch-wise conditional prior architecture. The input $x \in \mathbb{R}^{H \times W \times 3}$ is split into $N$ patches, and for each patch, a small CNN followed by an SSM layer outputs a single keypoint proposal. We filter top-$L$ prior keypoint proposals with the smallest sum of empirically-calculated SSM variance.

### D.3   DLPv2 - Encoder

The role of the encoder is to produce the posterior latent particles for an input image. To that end, it models the approximate posterior, $q_\phi(z|x) = q_\phi(z_a|x) \times q_\phi(z_o, z_s, z_d, z_t|x, z_a) \times q_\phi(z_f|x, z_p, z_s)$, where $z_a$ are anchor keypoints and $z_o$ are offsets from these anchors that together comprise $z_p = z_a + z_o$. The motivation for this modular approach originates in anchor-based object detection (Redmon et al., 2016), showing that using anchors produces more accurate bounding-boxes by relaxing the optimization objective to only predict the offset from the anchor instead of the exact position. This idea later propagated to unsupervised object detection (Lin et al., 2020c; Smirnov et al., 2021), significantly improving the detection performance. The encoding process is hierarchical and composed of 3 steps: (1) *anchor encoding* - a learned offset of $K$ keypoint

anchors from the $L$ keypoint proposals; (2) *attribute encoding* - extraction of position offset, scale and transparency attributes for each particle and (3) *appearance encoding* - encoding the particles visual features. In the following, we explain the design of each step.

We begin with encoding $K$ particle positions. DLP of Daniel & Tamar (2022a) uses a CNN followed by a fully-connected (FC) network to encode $K$ posterior keypoints *from the whole image*. We found this approach results in inaccurate object positions and misses small objects; in DLPv2, we depart from this approach to a fine-grained resolution approach where we utilize patches, or *glimpses*, $x_P \in \mathbb{R}^{L \times S \times S \times 3}$, where $S$ is a hyperparameter for the patch size. To that end, we model the *anchor encoder* $q_\phi(z_a|x)$ as follows: first, similarly to the keypoint proposals selection process described in Sec. D.2, we take the top-$K$ lowest SSM-variance keypoints out of the $L$ proposals to get a set of anchors $p_K$. Then, patches are extracted by using the prior keypoint proposals $p_L \in \mathbb{R}^{L \times 2}$ as input to a Spatial Transformer Network (STN, Jaderberg et al. 2015). Next, a small CNN outputs features $M(x_P)$ from each patch which are flattened and fed through a feedforward network (FFN) to output the offset $o(p)$ of each anchor which is added to the original $K$ selected anchors to get a set of *modified anchor* keypoints $z_a = p_K + o_K \in \mathbb{R}^{K \times 2}$. This set of keypoint anchors approximates objects location, and we proceed to the attribute encoding step where we lock on a more accurate position and extract the rest of the spatial attributes.

Next, to extract the position offset $z_o$, scale $z_s$, depth $z_d$ and transparency $z_t$ attributes for each particle, we extract glimpses from the anchor keypoints $z_a$ via STN as before, which are input to the *attribute encoder* $q_\phi(z_o, z_s, z_d, z_t | x, z_a)$ modeled as small CNN followed by a FC network that outputs the parameters of the distributions described in Section D.1 for each particle. Note that the network parameters are shared between all particles. In practice, the prior network is re-used as the *anchor encoder* $q_\phi(z_a|x)$ and the *attribute encoder* $q_\phi$ produces the offset from the anchors.

With all particle attributes in hand, we follow the original DLP and extract the particle visual features with the *appearance encoder* $q_\phi(z_f|x, z_p, z_s)$. Similarly to the previous steps, STN is used to extract glimpses of size $S$, with the one difference of using the scale $z_s$ as input to the STN in addition to $z_p = z_a + z_o$. The scale parameter is important when the regions around objects are smaller or larger than $S \times S$. Note that since we used STN in all steps, all parameters are differentiable end-to-end. The appearance encoder is designed similarly to the attribute encoder and outputs the Gaussian distribution parameters of the visual features surrounding the particle.

Finally, we assign an abstract particle for the background that is always located in the center of the image and its visual features $z_{\text{bg}}$ are encoded with a *background encoder* $q_\phi(z_{\text{bg}}|x, z_p)$ which is a CNN followed by a FC network, similarly to the original DLP. The input to the background encoder is a masked image – the posterior keypoints $z_p$ are used to generate $K$ masks of size $S \times S$ that mask out the corresponding regions of the original image $x$. The entire posterior encoding process is illustrated in Figure 17.

**Architecture details:** the *anchor encoder* first uses STN to extract patches from $K$ keypoints, $p_K \in \mathbb{R}^{K \times 2}$, that were filtered based on the keypoints-covariance out of the $L$ keypoint proposals. The patches are fed through a 3-layer, ReLU activated with 4-group Group Normalization, CNN with $[16, 32, 64]$ channels, where the final output of this CNN is flattened and fed through a 3-layer, ReLU activated, MLP with layer sizes $[256, 128, 2]$ to output *anchor offsets* $o_K \in \mathbb{R}^{K \times 2}$, which are activated with Tanh activation. Finally, the anchor encoder outputs the modified anchors $z_a = p_K + o_K \in \mathbb{R}^{K \times 2}$. Next, the *attribute encoder* operates similarly to the anchor encoder–patches are extracted from the modified anchors $z_a$ using STN and fed through similar CNN and MLP as before, with the only difference of linear output heads for each attribute, i.e., projection layers for the offset, scale, transparency and depth. Each head outputs the parameters of the approximate posterior distributions, that is, $\mu$ and $\log \sigma^2$ for the Gaussian attributes (offset, scale and depth) and $(\log a, \log b)$ for the Beta-distributed transparency attribute. We use log for numerical stability when calculating the KL-divergence during the optimization process. Finally, for the *appearance encoder*, we use again the same exact process with the same CNN-MLP architecture to encode the visual features of each particle. Differently from the previous encoding steps, this time we utilize the scale attribute as well as input to the STN to lock-on the object, and the dimension output head of the final MLP is the feature dimension $m$, which is a hyper-parameter. Typical values for $m$ are between $[3, 12]$. The *background encoder* is a 4-layer CNN with $[32, 64, 128, 256]$ or $[32, 32, 64, 64, 128]$ channels followed by a 3-layer MLP that models

the Gaussian distribution parameters of the background latent with layer sizes $[256, 256, 2m_{\text{bg}}]$, where $m_{\text{bg}}$ is the background latent feature dimension, which for simplicity, we set $m_{\text{bg}} = m$ in all our experiments.

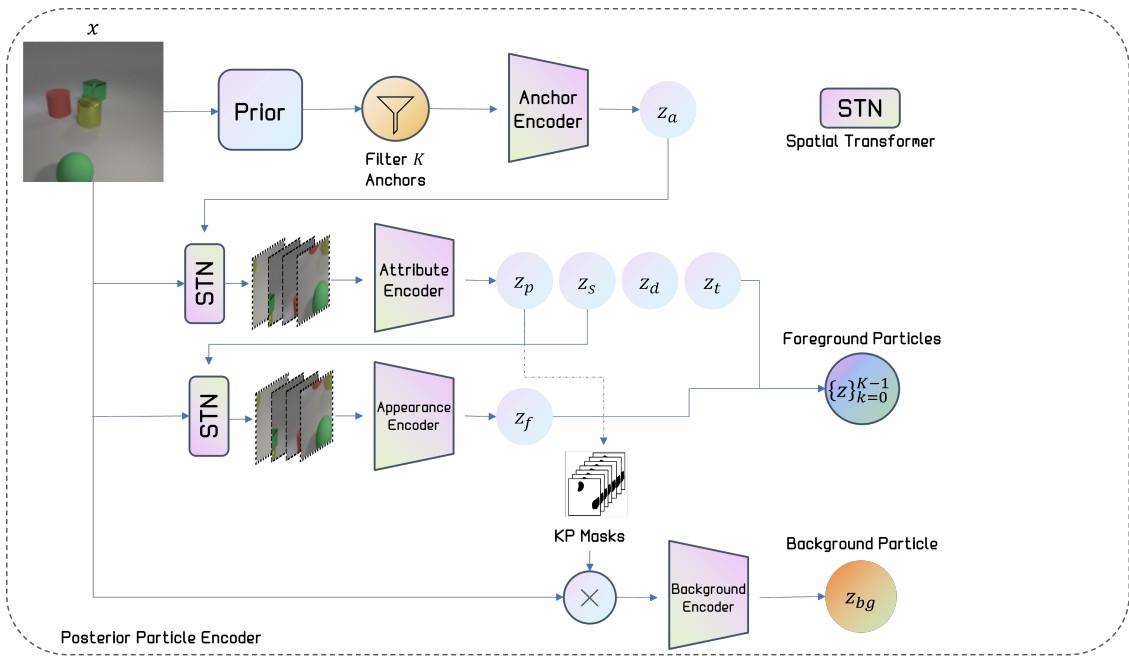

Figure 17: Posterior encoder architecture. We perform a hierarchical encoding process where first we approximate object locations by modifying the keypoint proposals that serve as anchors. Then, these anchors are used to extract precise attribute per particle, and finally, we encode the appearance features of each particle. The keypoints $z_p$ are used to generate masks to hide parts of the original image, which is then encoded with the background encoder.

### D.4 DLPv2 - Image Decoder

Recall that the decoder in DLP models the likelihood, $p_\theta(x|z) = p_\theta(x|z_p, z_s, z_d, z_t, z_{\text{bg}})$ and is composed of an *object decoder* and a *background decoder*. The object decoder follows the original DLP – a fully-connected layer followed by a small upsampling CNN that takes in a single latent particle features $z_f^{(i)}$, and decodes an RGBA patch $\tilde{x}_i^p \in \mathbb{R}^{S \times S \times 4}$, a reconstruction of each object's appearance. Then, the depth $z_s$ and transparency $z_t$ are used to factor the alpha channels of the object, which essentially determines the order of stitching and the visibility of the particle, and $z_p$ and $z_s$ are used to position and scale the decoded patches in the full $H \times W$ canvas, $\hat{x}_{\text{fg}}$, using a STN. The transparency and depth factorization process is detailed in a PyTorch-style code in Figure 18.

The background decoder is similar in design to standard VAEs, taking in $z_{\text{bg}}$ which is upsampled via a FC layer followed by a CNN to generate $\hat{x}_{\text{bg}}$. Finally, the foreground components and the background are stitched together by $\alpha \times \hat{x}_{\text{fg}} + (1 - \alpha) \times \hat{x}_{\text{bg}}$, where the masks $\alpha$ are the last channel of the RGBA glimpse output of the decoder. The decoder architecture is depicted in Figure 19.

**Architecture details:** the architecture of the *object decoder* mirrors the architecture of the appearance encoder: a 2-layer MLP takes in $z_f \in \mathbb{R}^m$ with 256 hidden units in each layer and outputs a feature vector which is reshaped to a 2D feature map that is then up-sampled by a CNN with similar reversed-order channels to the appearance encoder. The final CNN layer output a 4-channel 2D feature map activated with a Sigmoid activation, where the first channel is the decoded alpha map of the particle, and the rest are its RGB channels. The *background decoder* mirrors the background encoder, but in reverse, and outputs the RGB channels of the background component of the reconstructed image.

```
1   def factor_alpha_map(alpha_obj, rgb_obj, z_t, z_d):
2       # alpha_obj:[batch_size, num_particles, 1, h, w], glimpses alpha
3       # rgb_obj:[batch_size, num_particles, 3, h, w], glimpses RGB
4       # z_t:[batch_size, num_particles, 1], transparency attribute
5       # z_d:[batch_size, num_particles, 1], depth attribute
6
7       # factorize transparency
8       alpha_obj = alpha_obj * z_t
9       # [batch_size, num_particles, 1, h, w]
10      # factorize rgb - mask out irrelevant pixels
11      rgba_obj = alpha_obj * rgb_obg
12      # [batch_size, num_particles, 3, h, w]
13      # depth-based importance map - determine the stitching order
14      importance_map = alpha_obj * sigmoid(-z_d)
15      # [batch_size, num_particles, 1, h, w]
16      # normalize importance map between [0, 1]
17      importance_map = importance_map / (sum(importance_map, dim=1) + 1e-5)
18      # re-order and stitch the rgba maps
19      objects_canvas = (rgba_obj * importance_map).sum(dim=1)
20      # [batch_size, 3, h, w]
21      # create the mask for the background
22      bg_mask = 1 - (alpha_obj * importance_map).sum(dim=1)
23      # [batch_size, 1, h, w]
24      return objects_canvas, bg_mask
25
```

Figure 18: PyTorch-style implementation of the transparency and depth factorization process in the stitching process.

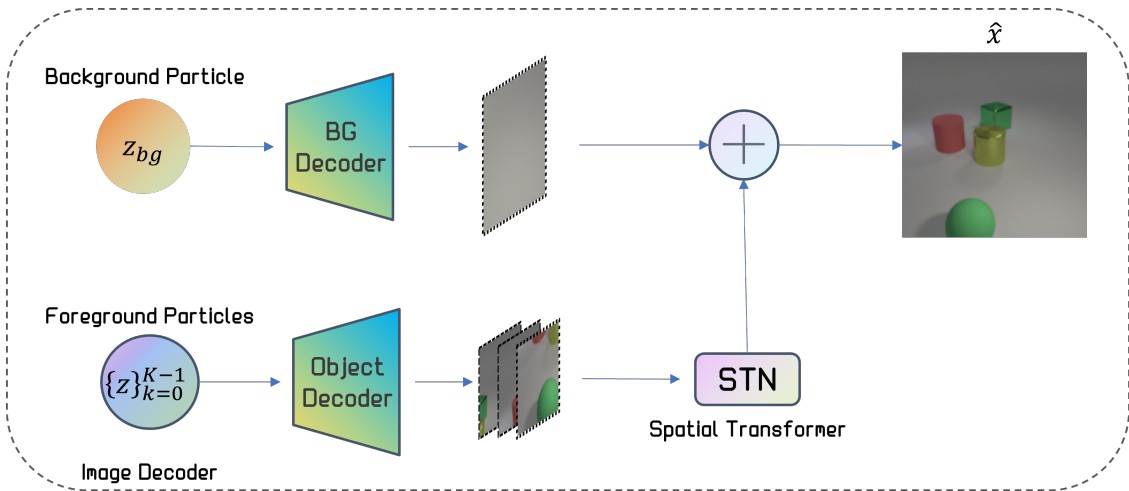

Figure 19: Image decoder architecture. The object decoder outputs an RGBA patch–a reconstruction of each object's appearance. The background decoder reconstructs the background component and finally, the foreground components and the background are stitched together to produce the final image reconstruction.

Up till now we have described DLPv2 for single images. We next continue to the details of the video prediction model DDLP.

## D.5 DDLP - Particle Tracking Posterior for Videos

Recall that for accurate dynamics modeling we must track the particles over time. Treating each frame individually would not work, as the order of the particles is not aligned between frames and there is no

guarantee that objects in consecutive frames will be assigned the same particle. The particle-alignment problem is illustrated in Figure 20.

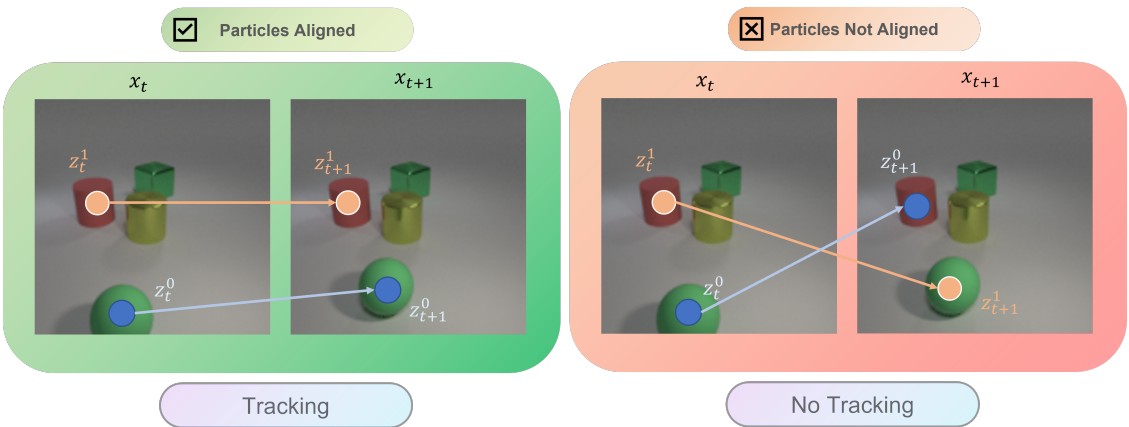

Figure 20: Illustration of the particle-alignment problem. Without tracking, i.e., treating each frame individually, there is no guarantee that the order of the particles is preserved between frames, e.g., particle #0 in frame $x_t$ (green ball) and particle #0 in frame $x_{t+1}$ (red cylinder) describe different objects.

To that end, we made an assumption that the displacement of objects between consecutive frames is small, such that the attribute encoder searches for the object in a small region around its previous location. We take 2 steps to implement particle tracking: (1) We use $z_{p,t-1}$ as the anchor keypoints for the next frame, i.e., $z_{a,t} = z_{p,t-1}$, where for $t = 0$ the anchors are extracted similarly to single-image setting (cf. Section 4.1) and (2) we compute the normalized cross-correlation between the extracted glimpses from $x^{t-1}$ and $x^t$ around $z_{p,t-1}$, implemented efficiently with group-convolution. In Figure 21 we provide a PyTorch-style code implementing the normalized cross-correlation. Finally, The single-channel generated score-map is concatenated channel-wise to the input of the attribute encoder.

```
1   def correlate(x, kernel):
2       # x: [batch_size, ch, h1, w1]
3       # kernel: [batch_size, ch, h2, w2]
4       batch_size, ch, h1, w1 = x.shape
5       x = x.reshape(1, batch_size * ch, h1, w1)
6       groups = batch_size  # for group-convolution
7       output = conv2d(x, kernel,
8                       groups=groups, padding=0, stride=1, bias=None)
9       # correlation identity to calculate the norm
10      corr_id = conv2d(x ** 2, ones_like(kernel),
11                      groups=groups, padding=0, stride=1, bias=None)
12      norm = sqrt(sum(kernel ** 2) * corr_id)
13      return output / (norm + 1e-5)
14
```

Figure 21: Efficient PyTorch implementation of normalized cross-correlation with group-convolution.

### D.6 DDLP - Generative Predictive Dynamics with Particle Interaction Transformer

The dynamics module models the prior distribution $p_\psi(z_{t+1}|z_t)$ of the particles temporal evolution. A good dynamics model should accurately capture the role of the particles' attributes and the interaction between the particles. For end-to-end training of the posterior and prior, previous VAE-based methods (Minderer et al., 2019; Lin et al., 2020b) used RNN-based dynamic modules as a stochastic prior, optionally combined with an interaction module (Battaglia et al., 2016). Alternatively, several approaches (Wu et al., 2022; 2021b) use a deterministic 2-step training scheme where in the first stage a representation of the visual states is

learned, and in the second stage a Transformer-based dynamic module models the sequence of latent variables where the attention mechanism is used to capture the relationship between entities.

Recall that our proposed dynamics module, *Particle Interaction Transformer (PINT)*, is a causal Transformer decoder based on the design of Generative Pre-trained Transformer (GPT, Radford et al. 2018). Differently from previous approaches, we do not add positional embeddings before the attention layers and use a learned per-head *relative positional bias* (Shaw et al., 2018; Raffel et al., 2020) which is directly added to the attention matrix before the softmax operation. We decompose the added relative positional bias to a *temporal* relative positional bias and a *spatial* relative bias. The relative positional bias is illustrated visually in Figure 23. $B_{\text{time}}$ and $B_{\text{pos}}$ are repeated $(K+1)$ and $N$ times respectively, and summed to create the relative positional bias matrix $B = B_{\text{time}} + B_{\text{pos}} \in \mathbb{R}^{((K+1)T) \times ((K+1)T)}$, $\{B\}_{ij} = b_i^j = b_i + b^j$, where $i$ denotes the relative time and $j$ the relative particle, as illustrated in Figure 22b. The relative positional bias is added directly to the query-key matrix, i.e., $\text{Attention} = \text{Softmax}\left(\frac{QK^T}{\sqrt{D}} + B\right)V$. To model particle interactions, the attention operation is modified such that particles attend to each other in the same time-step and across time, i.e. the attention matrix $A \in \mathbb{R}^{((K+1)T) \times ((K+1)T)}$, and similarly to GPT we use a causal mask to ensure the attention only considers past inputs, as illustrated in Figure 22a.

**Architecture details:** our particle interaction Transformer (PINT) models the stochastic prior for the particles and is constructed as follows: a 3-layer *projection MLP* with 256 hidden units takes in the latent particles' concatenated learned attributes $\{z\}_0^{T-2}$ and projects them to dimension $D = 256$. Next, a *particle Transformer*, based on the open-source minGPT (Karpathy, 2021) with 6 layers and 8 heads per layer for all datasets, performs attention between particles and across time as described before, where we directly add the relative positional bias to the attention matrix before softmax. Finally, a 3-layer *decoder MLP* outputs the prior distribution parameters with a similar architecture to the projection MLP. In practice, we model the residual change $\Delta = \text{PINT}(\{z\}_0^{T-2})$ of the particle attributes, i.e., $\{\hat{z}\}_1^{T-1} = \{z\}_0^{T-2} + \Delta$, for all attributes except for the transparency attribute which is directly modeled as PINT's output. To speed-up training, we use teacher forcing (Williams & Zipser, 1989) at training time (i.e., instead of feeding the dynamics module its previous prediction, the ground-truth value is used – the posterior particles in our case), while at inference time, predictions are generated autoregressively. We use GELU activations for all MLPs and the standard Layer Normalization in the MLPs and attention layers, where we use the pre-normalization architecture. We use Dropout regularization layers with $p = 0.1$ in the particle Transformer attention layers. We follow minGPT's initialization for the MLPs and attention layers where linear layers are initialized from a Gaussian distribution $\mathcal{N}(0, 0.02^2)$.

# E    Additional Training and Optimization Details

**Optimization objective:** the objective for DLPv2 follows the original DLP. Given an input image $x \in \mathbb{R}^{H \times W \times 3}$, we optimize:

$$\mathcal{L}_{\text{DLP}} = \mathcal{L}_{rec}(x, \tilde{x}) + \beta_{\text{KL}}\left(\text{ChamferKL}(q_\phi(z_p|x)\|p_\gamma(z_p|x)) + \text{KL}(q_\phi(z_{a|f}|x)\|p(z_{a|f})) + \beta_f \text{KL}(q_\phi(z_f|x)\|p(z_f))\right), \quad (2)$$

where $\tilde{x}$ is the reconstructed image, $\mathcal{L}_{rec}(x, \tilde{x})$ is the reconstruction loss, $z_p$ are the posterior keypoints, $p_\gamma(z_p|x)$ denotes the patch-wise spatial-softmax prior of the original DLP, $z_{a|f}$ are all the attributes–scale, transparency and depth, except for the appearance features $z_f$, and $\beta_{\text{KL}}$ and $\beta_f$ are coefficients to balance the reconstruction loss and the KL-divergence terms. Note that the effective coefficient for the visual features is $\beta_{\text{KL}} \cdot \beta_f$, and in all our experiments we set $\beta_f = 0.001$ following the original DLP (Daniel & Tamar, 2022a).

For DDLP, we optimize the sum of ELBOs over time, $\mathcal{L}_{\text{DDLP}} = -\sum_{t=0}^{T-1} \text{ELBO}(x_t) = \sum_{t=0}^{\tau-1} \mathcal{L}_{\text{DLP},t} + \sum_{t=\tau}^{T-1} \mathcal{L}_{\text{dyn},t}$, where $\mathcal{L}_{\text{DLP},t}$ is defined in Equation 2 and

$$\mathcal{L}_{\text{dyn},t} = \mathcal{L}_{rec}(x_t, \tilde{x}_t) + \beta_{\text{KL}}\text{KL}(q_\phi(z_t|x)\|p_\psi(z_t|z_{t-1})),$$

where here we use the same $\beta_{\text{KL}}$ coefficient as before for *all temporal attributes*. As the prior parameters for the KL-divergence are the output of PINT, and not hyper-parameter constants, we found that there was no need, in terms of performance, to explicitly introduce separate coefficients for the temporal attributes, though further investigation may be done to validate this.

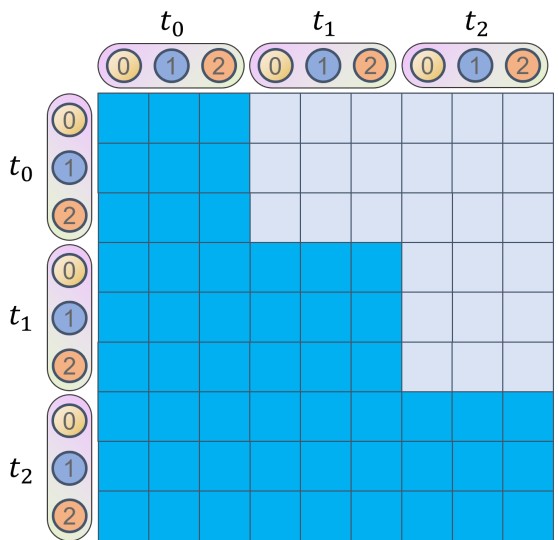
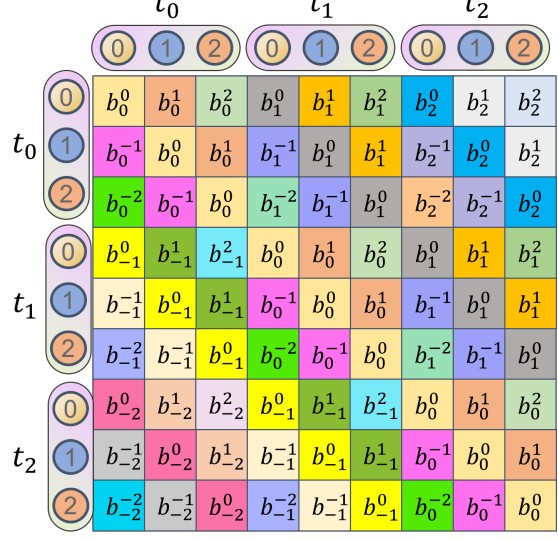

(a) Attention in PINT. Modified to allow particles to attend to each other. A causal mask is applied to attend past and current particles.

(b) Relative positional bias matrix in PINT. $b_i^j = b_i + b_j$, where $i$ denotes the relative time and $j$ the relative particle position.

Figure 22: Particle Interaction Transformer (PINT): attention and positional bias.

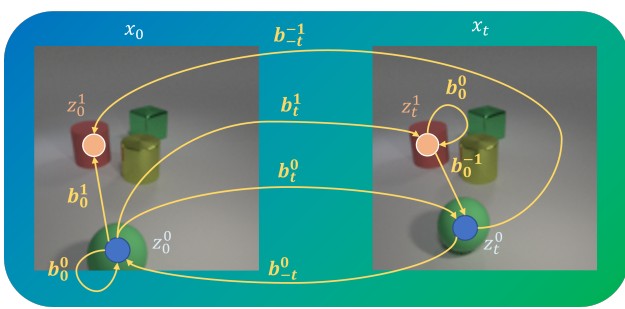

Figure 23: Illustration of the relative positional encoding. Particles in the same time-step $t$ get the same temporal relative embedding $b_t$, while across time, particle $i$ get a unique relative positional embedding w.r.t the other particles $b^i$. Together, each particle is assigned a space-time relative positional encoding $b_t^i$.

**Closed-form formula for Beta distribution:** recall that the transparency attributes are modeled as Beta distributed random variables, which requires us to calculate the KL-divergence between the posterior and prior parameters. Fortunately, similar to the Gaussian distribution, the KL-divergence for the Beta distribution has a closed-form formula. Let $X_1 \sim \text{Beta}(a, b)$ and $X_2 \sim \text{Beta}(a', b')$, then:

$$\mathcal{D}_{\text{KL}}(X_1 || X_2) = \ln\left(\frac{B(a', b')}{B(a, b)}\right) + (a - a')\psi(a) + (b - b')\psi(b) + (a' - a + b - b')\psi(a + b),$$

where $B(x, y) = \frac{\Gamma(x)\Gamma(y)}{\Gamma(x+y)}$ and $\Gamma$ is the Gamma function, and $\psi = \frac{d}{dx}\ln(\Gamma(x)) = \frac{\Gamma'(x)}{\Gamma(x)}$ is the logarithmic derivative of the Gamma function ("digamma"). In PyTorch, the closed-form formula can be implemented using:

```
1    # logarithm of the gamma function
```

```
2          log_gamma = torch.lgamma(x)
3          # logarithmic derivative of the gamma function
4          psi = torch.digamma(x)
```

**Separable Chamfer-KL:** in practice, we found that for DLPv2, keypoint positions lock more accurately on the center of objects if we model the anchors $z_a$ as deterministic variables instead of random variables, and only the offset from the anchor $z_o$ is a random Gaussian variable, making $z_p = z_a + z_o$ a random Gaussian variable as well. Implementation-wise, we optimize the deterministic Chamfer distance between the posterior anchor keypoints and the proposal prior keypoints, where for the offset we optimize the standard Gaussian KL-divergence. In DDLP, the above is relevant only for the first $t < \tau$ frames, as they are optimized w.r.t the constant prior parameters as described in the main text. For $t \geq \tau$, we use the standard Gaussian KL-divergence between the posterior keypoints and PINT's output prior parameters.

**Optimization:** following is a description of finer training details; first, similarly to the original DLP, we perform a warm-up stage where the background encoder and decoder are not trained to give an advantage to the object modeling components. We reserve the first epoch for the warm-up, except for the Traffic dataset, which due its smaller size required 3 warm-up epochs. After the warm-up step, we reserve one epoch for a noisy step where the learned alpha channels are artificially added a small Gaussian noise sampled from $\mathcal{N}(0, 0.1^2)$ to learn sharper masks. In addition, we gradually anneal the weight of $\mathcal{L}_{\text{dyn}}$ from 0 to 1 at a linear rate over the first 10,000 iterations for a more stabilized training of the Transformer.

**Hyper-parameters:** all CNNs are initialized from a small Gaussian distribution $\mathcal{N}(0, 0.01^2)$ and use replication-padding. We use the Adam (Kingma & Ba, 2014) optimizer ($\beta_1 = 0.9, \beta_2 = 0.999, \epsilon = 1e - 4$) with initial learning rate of $2e - 4$ and a step scheduler that multiplies the learning rate by 0.95 at the end of each epoch. The constant prior distribution parameters are the same for all datasets and reported in Table 5. The complete set of the rest of the hyper-parameters can be found in Table 4.

| Dataset | Balls-Interaction | OBJ3D / CLEVRER | Traffic | PHYRE |
|---|---|---|---|---|
| Input Frames (Train/Inference) $T$ | 20/10 | 10/6 | 10/6 | 15/10 |
| Posterior KP $K$ | 10 | 12 | 25 | 25 |
| Prior KP Proposals $L$ | 16 | 64 | 64 | 64 |
| Reconstruction Loss | MSE | Perceptual | Perceptual | MSE |
| $\beta_{KL}$ | 0.1 | 40 | 40 | 0.1 |
| Prior Patch Size | 8 | 16 | 16 | 16 |
| Glimpse Size $S$ | 16 | 32 | 32 | 16 |
| Feature Dim $d$ | 3 | 8 | 8 | 4 |
| Epochs | 20 | 30 | 25 | 40 |

Table 4: Detailed hyperparameters used for the various experiments in the paper.

| Attribute | Distribution | Parameters |
|---|---|---|
| Position Offset $z_o$ | Normal, $\mathcal{N}(\mu, \sigma^2)$ | $\mu = 0, \sigma = 1$ |
| Scale $z_s$ | Normal, $\mathcal{N}(\mu, \sigma^2)$ | $\mu = \text{Sigmoid}^{-1}\left(\frac{\text{glimpse size}}{\text{image size}}\right), \sigma = 1$ |
| Depth $z_d$ | Normal, $\mathcal{N}(\mu, \sigma^2)$ | $\mu = 0, \sigma = 1$ |
| Transparency $z_t$ | Beta, $\text{Beta}(a, b)$ | $a = 0.1, b = 0.1$ |
| Appearance Features $z_f$ | Normal, $\mathcal{N}(\mu, \sigma^2)$ | $\mu = 0, \sigma = 1$ |

Table 5: Prior Distribution Parameters

# F  Complexity Analysis

Our experiments with DDLP and G-SWM were conducted primarily on machines equipped with 4 NVIDIA RTX 2080 11GB GPUs or 1 NVIDIA A4000 16GB GPU. We trained SlotFormer on 4 NVIDIA A100 80GB GPUs. For the evaluated models, we provide a summary of several complexity parameters in Table 6. We

used the `OBJ3D` dataset at $128 \times 128$ resolution for our evaluation setting. The inference time was calculated based on SlotFormer's evaluation setting, which involves 6 conditional frames and unrolling to 50 frames. The reported values are for the models used in our paper's results, with G-SWM and SlotFormer trained using recommended hyperparameters. We report the inference time on a standard CPU and inference on an A4000 GPU. Note that DDLP requires significantly less memory compared to SlotFormer, which has twice the number of slots than our particles, even though both use transformer-based dynamics modules. Typical training of DDLP takes up to 3 days on 4 NVIDIA RTX 2080 11GB GPUs for $128 \times 128$ resolution datasets, and up to 1.5 days on similar hardware for $64 \times 64$ resolution datasets.

| Model | Model Size | GPU Memory (per GPU) | Inference Time [ms] |
|---|---|---|---|
| DDLP $K = 12$ | 10M ( 40MB) | 9GB | CPU: $1234 \pm 25.7$, GPU: $250 \pm 2.8$ |
| SlotFormer $K = 6$ | 4M ( 15MB) | Stage 1: 10GB, Stage 2: 22 GB (+FP16) | CPU: $6923 \pm 289.3$, GPU: $222 \pm 12.3$ |
| G-SWM | 13M ( 50MB) | 13GB | CPU: $314 \pm 17.6$, GPU: $213 \pm 11.7$ |

Table 6: Model size, GPU memory usage and inference time for different models. $K$ is the number of particles/slots.

# G   Datasets

We provide extended details on the datasets used throughout this paper. The datasets can be roughly categorized to (1) *dense interactions*, where interactions between objects occur frequently and most of the sampled sequences include interactions, and (2) *sparse interactions*, where interactions are less frequent or occur in the distant horizon, and sampled sequences may contain no interactions at all.

`Balls-Interaction` a dense interaction 2D dataset introduced in Jiang et al. (2019) consists of videos of 3 random-colored balls bouncing and colliding with each other. Each video, or episode, is 100-frame long, where each frame is of size $64 \times 64$. We follow Lin et al. (2020b) generation process and use 10,000 episodes for training, 200 for validation and 200 for test.

`OBJ3D` a dense interaction 3D dataset containing CLEVR-like objects (Johnson et al., 2017) introduced in Lin et al. (2020b). In each 100-frame video of $128 \times 128$ resolution, a random-colored ball is rolled towards a random number of objects of different types positioned in the center of the scene, colliding with them. The dataset contains 2,920 episodes for training, 200 for validation and 200 for test.

`CLEVRER` a sparse interaction 3D dataset containing CLEVR-like objects introduced in Yi et al. (2019). Each video consists of objects entering the scene from various locations, potentially colliding. As new objects may enter the scene at future times-teps, we trim every video to maximize the visibility of objects in the initial scene and we resize the frames to $128 \times 128$. We use the the first 5,000 videos for training, 1,000 for validation and 1,000 for test.

`PHYRE` a sparse interaction 2D dataset of physical puzzles introduced in Bakhtin et al. (2019), designed for physical reasoning. We collect simulated data in the form of $128 \times 128$ frames from the BALL-tier tasks in the ball-within-template setting, where tasks are considered solved if the user-placed ball meets a certain condition, e.g., touch the wall/floor/another object. We generate rollouts from all tasks, except for tasks $[12, 13, 16, 20, 21]$ which include a lot of distractions. We use 2,574 episodes for training, 312 for validation and 400 for test.

`Traffic` a real-world dataset introduced in Daniel & Tamar (2022a) containing videos of cars of different sizes and shapes driving along opposing lanes, captured by a traffic camera. We separate the video to episodes of 50 frames, each frame of size $128 \times 128$. The dataset contains 133,000 frames, where we take 80% for training, 10% for validation and 10% for test.

# H   Extended Comparison with SlotFormer

In this section, we present an extended comparison between our proposed DDLP model and SlotFormer, a recent slot-based video prediction model that achieves state-of-the-art results (Wu et al., 2022). To conduct our experiments, we used the available official code and pre-trained models (Wu et al., 2023). It is important

to note that the results reported in (Wu et al., 2022) are on the validation set – the same dataset that was used for tuning hyperparameters, and are for the $64 \times 64$ resolution version of `OBJ3D`. To ensure consistency with our paper, we used their pre-trained model to evaluate the reported metrics on the validation set and also computed the same metrics on a test set (which was not used for training nor hyperparameter tuning). Our paper only reports results for the test sets. For a visual comparison between DDLP and SlotFormer, please refer to the videos available at our webpage: `https://taldatech.github.io/ddlp-web/`.

`OBJ3D` comparison: we compared DDLP and SlotFormer on the `OBJ3D` dataset in two experiments. First, we trained a $64 \times 64$ version of DDLP to compare with SlotFormer's pre-trained model in their evaluation setting. Second, we trained a $128 \times 128$ version of SlotFormer and compared it to our model. We evaluated both experiments in SlotFormer's evaluation setting, which involves 6 conditional frames and a rollout horizon of 50 for a fair comparison. Based on the LPIPS metric, which is better aligned with human perception (Wu et al., 2022), our model outperforms SlotFormer, as shown in Table 7. Figure 24 and the video rollouts on the webpage demonstrate that SlotFormer struggles when objects rotate, resulting in object deformation.

| | `OBJ3D` $64 \times 64$ | | | `OBJ3D` $128 \times 128$ | | |
|---|---|---|---|---|---|---|
| | PSNR ↑ | SSIM ↑ | LPIPS ↓ | PSNR ↑ | SSIM ↑ | LPIPS ↓ |
| **SlotFormer** | $32.16 \pm 5.43$ | $0.90 \pm 0.07$ | $0.08 \pm 0.06$ | $31.2 \pm 4.9$ | $0.92 \pm 0.04$ | $0.135 \pm 0.05$ |
| **DDLP (Ours)** | $32.01 \pm 5.42$ | $0.90 \pm 0.07$ | $0.07 \pm 0.06$ | $30.1 \pm 4.86$ | $0.91 \pm 0.04$ | $\mathbf{0.088 \pm 0.06}$ |

Table 7: DDLP vs. SlotFormer video prediction results on the `OBJ3D` dataset. For $64 \times 64$ resolution, we evaluate the publicly available pre-trained model. Results are reported on the test-set.

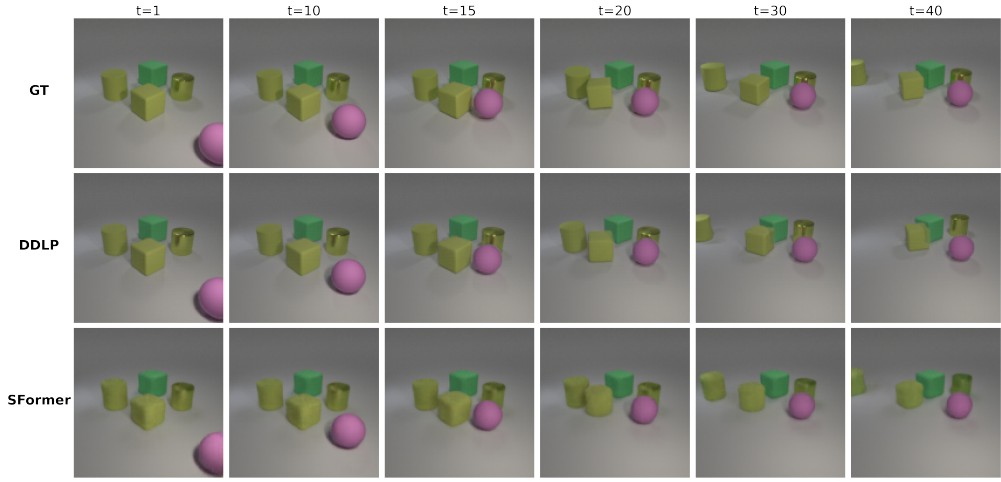

Figure 24: DDLP vs. SlotFormer video prediction comparison on the `OBJ3D` dataset.

`Traffic` comparison: for the `Traffic` dataset, as it includes many moving cars, we assumed that a large number of slots is required. SlotFormer used a maximum number of 8 slots in their experiments. We tried training it with 10 slots, but faced GPU memory issues on the same hardware required to train a $128 \times 128$ `Traffic` model; however, we were able to train SlotFormer with 10 slots on stronger cloud hardware. We attempted several runs until we got it to work, see our note regarding instability below. Since the `Traffic` dataset has scenes with more than 10 cars, our underlying assumption is that each slot can model more than 1 car. This assumption is reasonable, as the dynamics of the dataset are not complex, with cars moving in the same direction per lane, and the latent dimension of each slot is high (128), much larger than the dimension of each particle in DDLP. As shown in Figure 10, SlotFormer's long-horizon predictions become blurry. We provide more visual results of SlotFormer on the `Traffic` dataset for $64 \times 64$ and $128 \times 128$ resolutions on the webpage. Our visualization of per-slot rollouts, available on the webpage due to their visual size, indeed validates that slots sometimes model more than a single object, affecting the predictions when rolling out the dynamics model. Clearly, DDLP outperforms SlotFormer in this domain.

To further explain these results, recent studies into slot-based models attempt to shed light on why they struggle with real-world datasets. As Seitzer et al. (2023) concluded, image reconstruction is not enough for slot-based decomposition, and feature-wise loss from a pre-trained self-supervised method can mitigate the limitations of slot attention. Specifically, Seitzer et al. (2023) shows that the slot-based models STEVE (Singh et al., 2022b) and SAVi++ (Elsayed et al., 2022), which are used in SlotFormer, are mostly limited to synthetic datasets. Replacing these methods with the one proposed in Seitzer et al. (2023) might improve SlotFormer, although it would increase the training complexity.

Furthermore, while SlotFormer was evaluated on the `CLEVRER` and `PHYRE` datasets, we found it challenging to compare DDLP with SlotFormer as Wu et al. (2022) reported special pre-processing steps required for SlotFormer to work on these datasets. For `PHYRE`, the base slot-attention was reported to fail to detect objects in light colors, resulting in poor segmentation results that were addressed by manually adjusting specific pixel values. While this was not necessary for DDLP to work on this dataset, we used a color-inverted version of `PHYRE` for visual clarity, without any manual pixel adjustments. For `CLEVRER`, Wu et al. (2022) trained on $64 \times 64$ resolution images, subsampled each episode by a factor of 2. In contrast, we trained DDLP on $128 \times 128$ resolution images without subsampling. Note that both DDLP and SlotFormer cannot handles new objects that appear in the scene. To account for this, Wu et al. (2022) manually filtered out video clips where new objects appeared during the rollout period. We did not manually filter clips. To handle object appearance, we noticed that trimming a fixed number of frames from the beginning and end of the clips yielded clips with much less objects that enter the scene.

We encountered several challenges when training SlotFormer. Firstly, the hardware requirements for SlotFormer were higher than those for DDLP. We found that the machines we used to train DDLP (4xRTX2080) were not sufficient to train SlotFormer, which is in line with the reported requirements in the official repository (Wu et al., 2023). Additionally, as SlotFormer is trained in two stages, each with different memory requirements, we had to resort to using cloud-compute (4 NVIDIA A100 80GB GPUs) to run the experiments, making it less accessible than our model. For a complexity analysis, please see Appendix F, and for a comparison of complexity between DDLP and SlotFormer, please see Table 6. Secondly, we encountered instability during the first stage of SlotFormer's training, which involves training the base SAVi++ slot-attention model and required several attempts to make it work on the `Traffic` dataset. This aligns with the official code repository which recommends running 3 different runs with the same hyper-parameters and picking the best one. In our experience, DDLP was more stable to train and search for working hyper-parameters.

# I   Additional Results

This section presents additional visual results from the datasets we experimented on. The complete evaluation metrics for video prediction on all datasets are reported in Table 9.

## I.1   DLPv2 - Unsupervised Object-centric Scene Decomposition

Here we detail our improved DLPv2 scene decomposition capabilities. In Figure 8 we demonstrate how a single image is decomposed to particles by plotting the keypoints, objects' bounding boxes with confidence-based scores, objects' learned masks and the learned background. The bounding-box scores are calculated based on the the per-particle uncertainty, similarly to the original DLP:

$$V(u_k) \doteq \sum_i \log(\sigma_{k_i}^2),$$

where $\sigma_{k_i}$ is the standard deviation in the $i^{th}$ axis (i.e., the $x$ and $y$ coordinates) of $u_k$. In addition, when integrating the tracking posterior, we are able to track the particles through time, as demonstrated in Figure 25.

### I.1.1   Comparison of DLPv2 vs. DLPv1

In this section, we quantitatively and qualitatively compare our proposed method, DLPv2, with the original DLP using the publicly available implementation (Daniel & Tamar, 2022b). We train both models in the

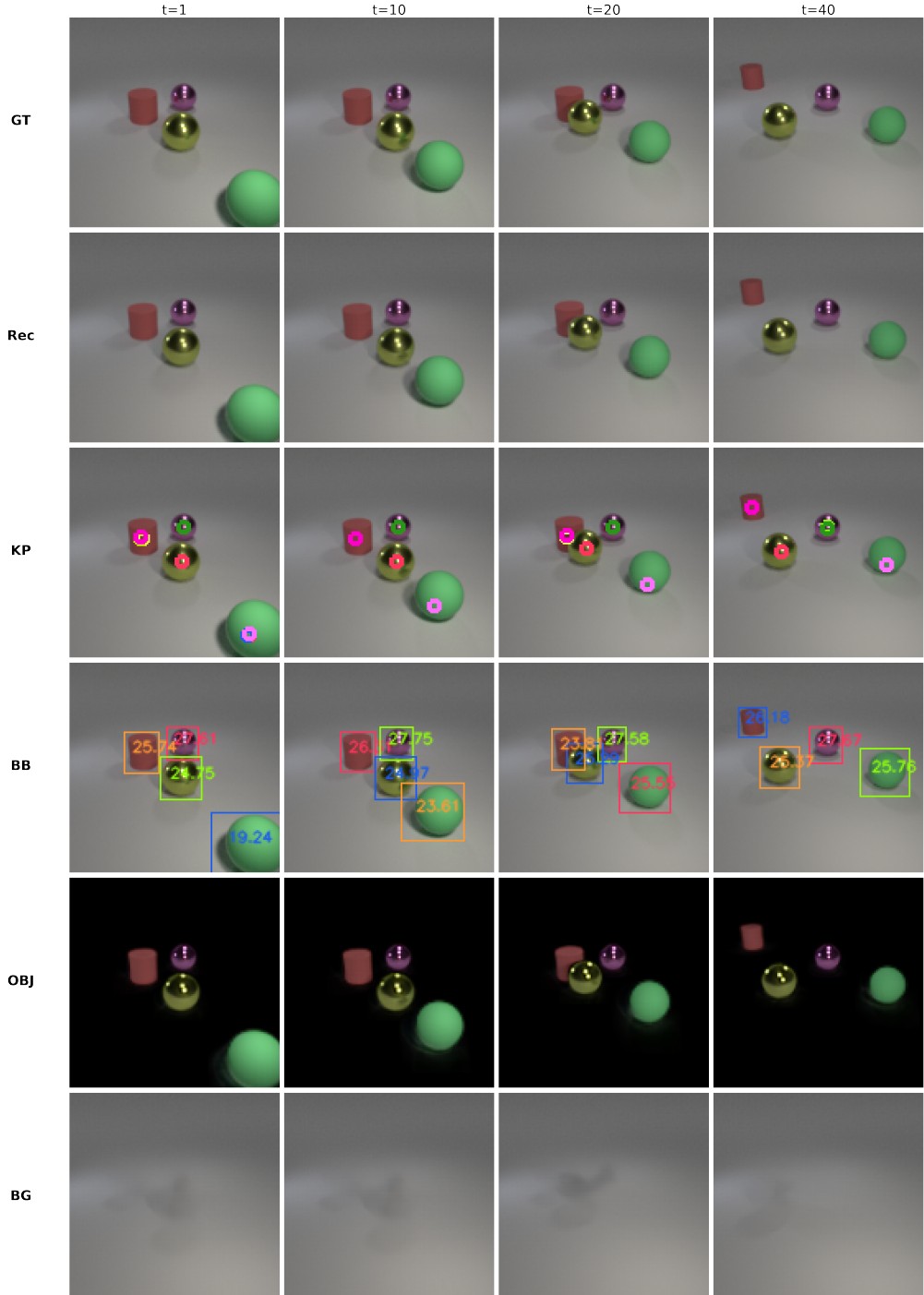

Figure 25: Tracking particles through time. We plot the learned keypoints, bounding boxes and objects. Notice how the order of the particles is preserved over time.

single-image setting on the `Traffic` and `OBJ3D` datasets, using the same number of particles and recommended hyper-parameters. We stop training once the validation LPIPS metric stops improving. As shown in Table 8, our proposed DLPv2 significantly outperforms the original DLP, indicating that our introduced modifications capture objects more accurately, resulting in improved image reconstructions. Qualitative image decomposition

comparisons between the two models are presented in Figures 26 and 27. Notably, DLPv2 generates more accurate particle positions, which are located at the center of objects, resulting in a more disentangled representation. Furthermore, it should be noted that DLPv1 does not model object attributes and, as a result, is incapable of generating bounding boxes.

| | OBJ3D | | | Traffic | | |
|---|---|---|---|---|---|---|
| | PSNR ↑ | SSIM ↑ | LPIPS ↓ | PSNR ↑ | SSIM ↑ | LPIPS ↓ |
| **DLP** | 39.23 ± 3.33 | 0.982 ± 0.009 | 0.085 ± 0.018 | 24.21 ± 0.9 | 0.67 ± 0.05 | 0.13 ± 0.02 |
| **DLPv2 (Ours)** | **41.97 ± 3.74** | **0.985 ± 0.006** | **0.019 ± 0.01** | **25.34 ± 0.8** | 0.67 ± 0.05 | 0.13 ± 0.01 |

Table 8: DLP vs. DLPv2 image reconstruction comparison in the single-image setting. Results are reported on the test-set.

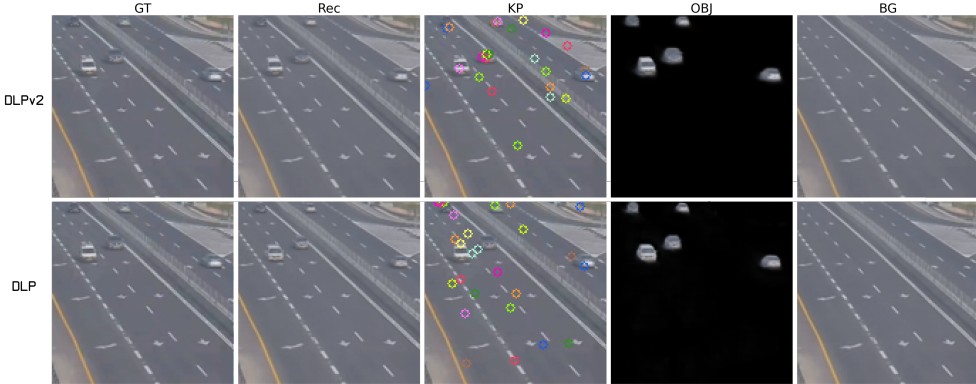

Figure 26: DLP vs. DLPv2 single-image decomposition on the `Traffic` dataset. It can be seen that DLPv2 produces more accurate particle positions that are located in the centers of the cars.

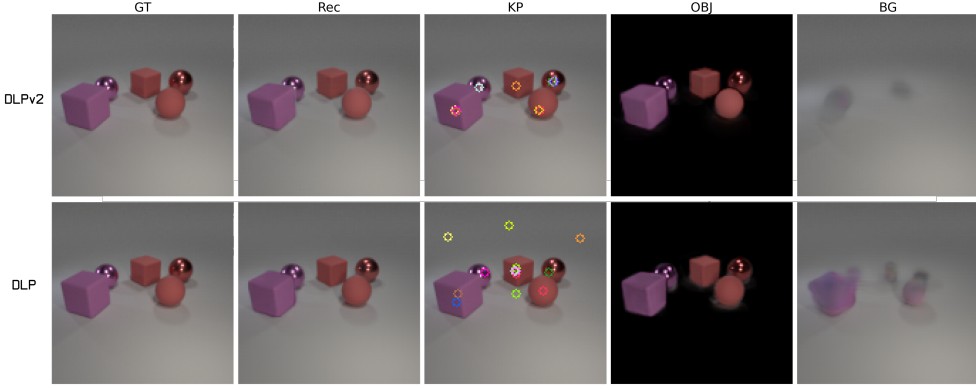

Figure 27: DLP vs. DLPv2 single-image decomposition on the `OBJ3D` dataset. It can be seen that DLPv2 produces more accurate particle positions that are located in the center of the objects, leading to a better foreground-background segmentation.

## I.2 Video Prediction: Balls-Interaction

Figures 28 and 29 present rollouts from the `Balls-Interaction` dataset, which contains a lot of collisions. The models are conditioned on the first 10 frames and predict 90 frames into the future. Note how the accumulated error of the balls position of G-SWM is larger than DDLP's predictions as also demonstrated quantitatively in Table 10.

| Dataset | OBJ3D | | | Traffic | | |
|---|---|---|---|---|---|---|
| | PSNR ↑ | SSIM ↑ | LPIPS ↓ | PSNR ↑ | SSIM ↑ | LPIPS ↓ |
| **G-SWM** | 31.7±6.2 | 0.924±0.05 | 0.118±0.07 | 24.88±1.07 | 0.58±0.07 | 0.235±0.04 |
| **SlotFormer** | 31.2±4.91 | 0.925±0.04 | 0.135±0.05 | 24.93±0.85 | 0.60±0.04 | 0.18±0.03 |
| **DDLP (Ours)** | 31.29±5.22 | 0.923±0.04 | **0.088±0.06** | 24.4±1.02 | 0.61±0.06 | **0.15±0.02** |
| Dataset | CLEVRER | | | PHYRE | | |
| **G-SWM** | 30.12±6.69 | 0.93±0.04 | 0.146±0.07 | 24.64±6.25 | 0.93±0.05 | 0.078±0.06 |
| **DDLP (Ours)** | 29.21±6.16 | 0.92±0.04 | 0.134±0.08 | 26.98±5.3 | 0.95±0.04 | **0.055±0.04** |

Table 9: Quantitative results on video prediction.

| Balls-Interaction | MED10 ↓ | PSNR ↑ | SSIM ↑ | LPIPS ↓ |
|---|---|---|---|---|
| **G-SWM** | 0.256 | 13.86±5.5 | 0.753±0.12 | 0.232±0.15 |
| **DDLP (Ours)** | **0.149** | 16.74±8.3 | **0.789±0.12** | **0.189±0.15** |

Table 10: `Balls-Interaction` dataset comparison. MED10 is the mean euclidean distance error of predicted balls positions, summed over the first 10 steps of generation.

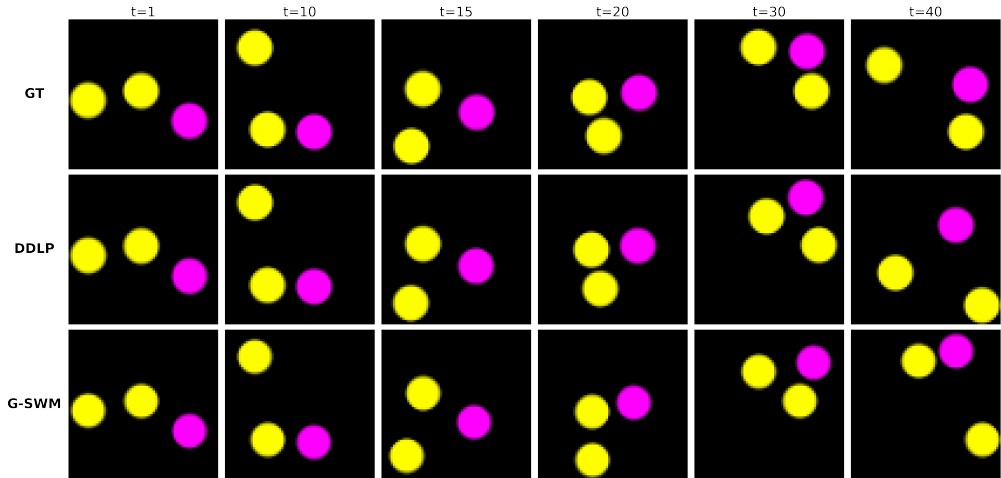

Figure 28: Rollout from the `Balls-Interaction` dataset.

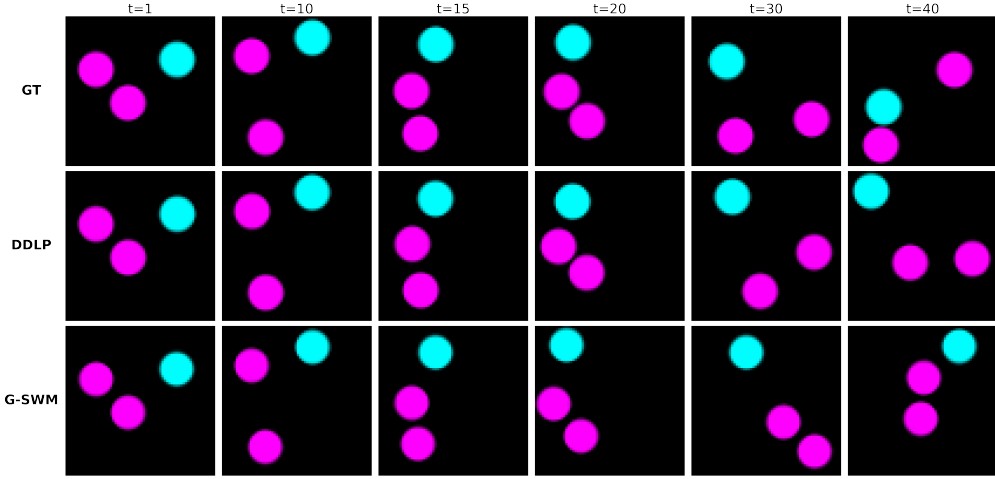

Figure 29: Rollout from the `Balls-Interaction` dataset.

### I.3 Video Prediction: Traffic

Figures 30 and 31 present more rollouts from G-SWM and DDLP trained on the `Traffic` dataset. The models are conditioned on the first 10 frames and predict 40 frames into the future. Note how cars in G-SWM's rollouts get blurry or vanish in the long term.

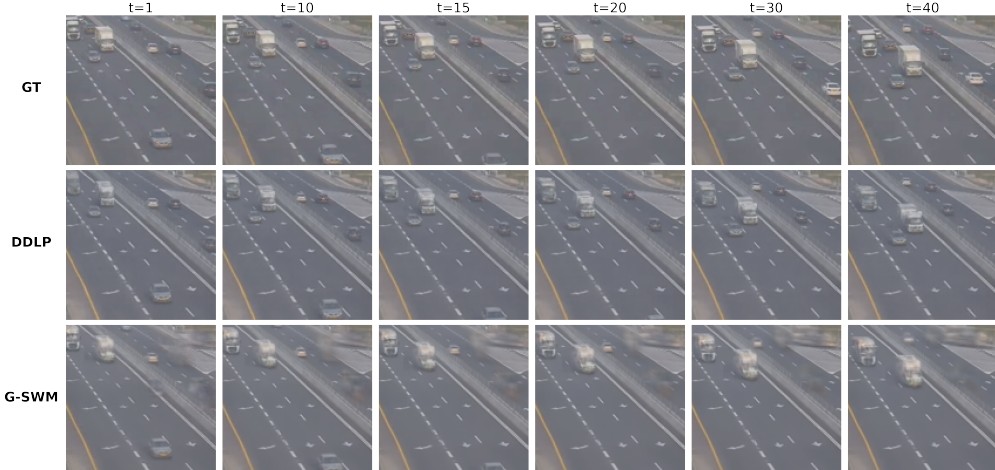

Figure 30: Rollout from the `Traffic` dataset.

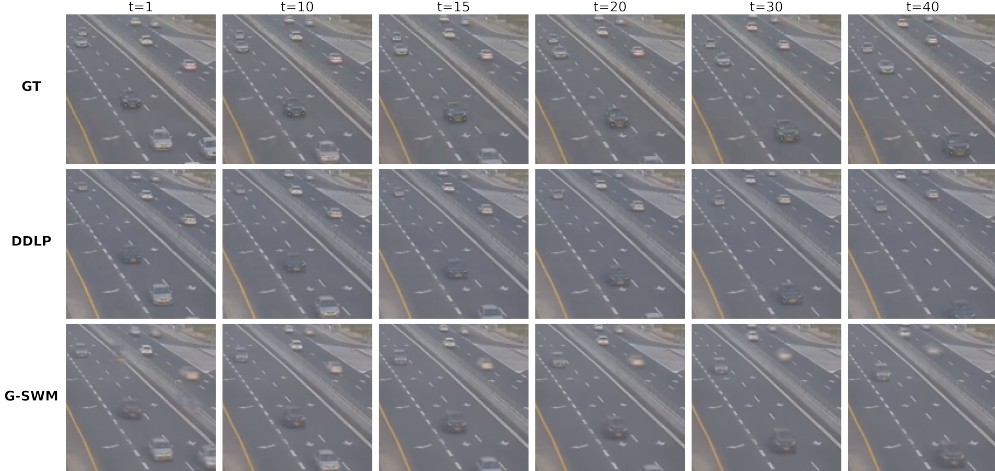

Figure 31: Rollout from the `Traffic` dataset.

### I.4 Video Prediction: OBJ3D

Figures 32 and 33 present more rollouts from G-SWM and DDLP trained on the `OBJ3D` dataset. The models are conditioned on the first 10 frames and predict 90 frames into the future. Note how objects in G-SWM's rollouts get blurry in the long term.

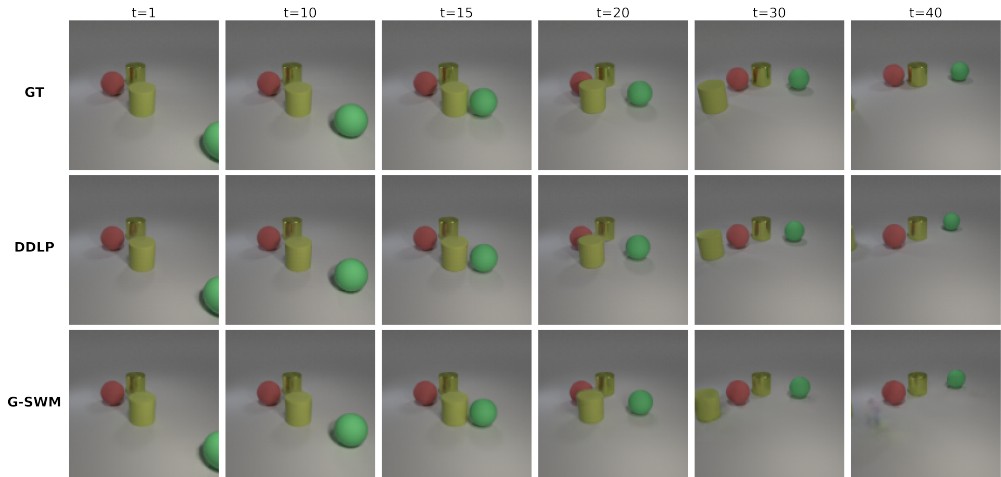

Figure 32: Rollout from the `OBJ3D` dataset.

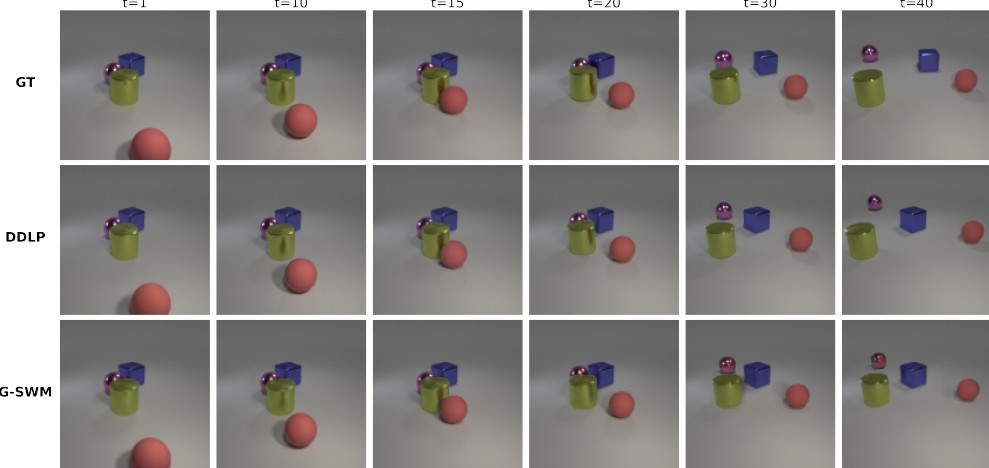

Figure 33: Rollout from the `OBJ3D` dataset.

### I.5 Video Prediction: PHYRE

Figures 34, 35, and 36 present rollouts from G-SWM and DDLP trained on the `PHYRE` dataset. The models are conditioned on the first 10 frames and predict 40 frames into the future. Note how objects in G-SWM's rollouts get blurry or vanish in the long term.

### I.6 Video Prediction: CLEVRER

Figures 37 and 38 present rollouts from G-SWM and DDLP trained on the `CLEVRER` dataset. The models are conditioned on the first 10 frames and predict 90 frames into the future.

### I.7 What If...? Video Prediction

In Figures 39, 40 and 41 we visualize more examples of latent particle modification of videos from `OBJ3D`.

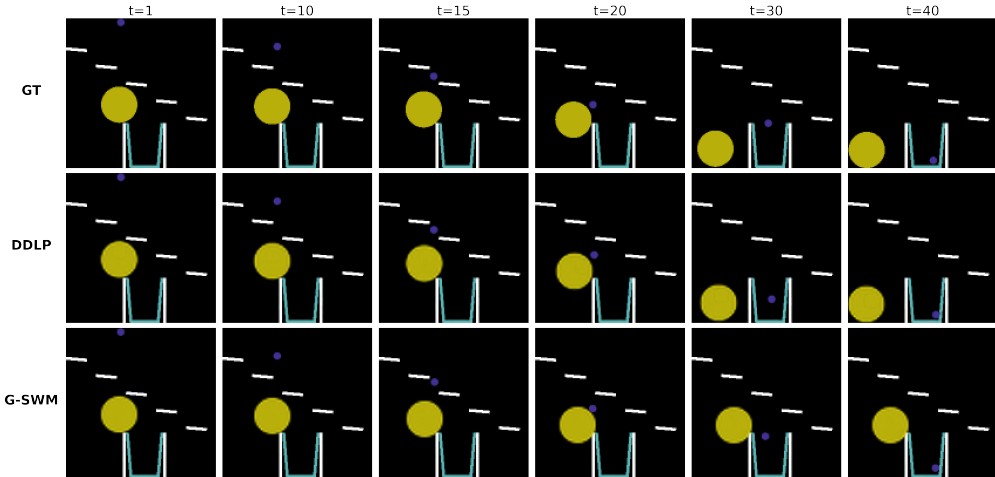

Figure 34: Rollout from the `PHYRE` dataset.

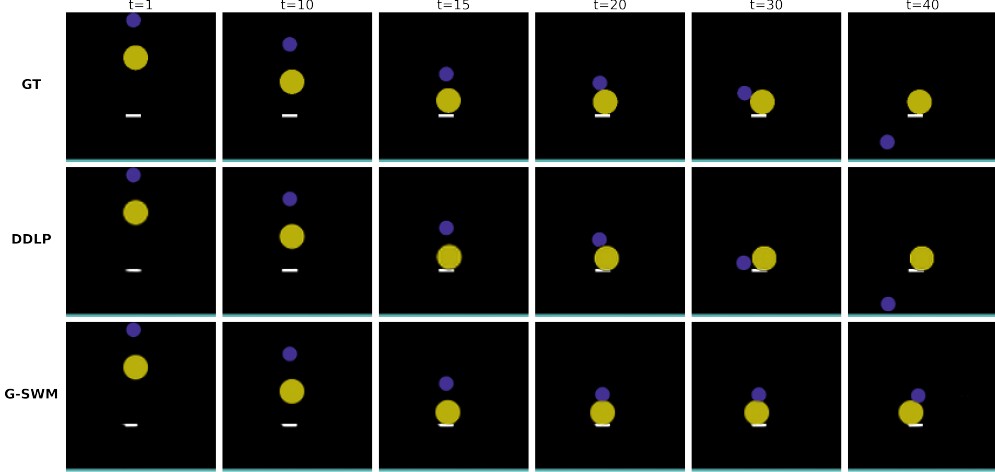

Figure 35: Rollout from the `PHYRE` dataset.

### I.8   Ablation Study

We evaluate the contribution of the design choices of DLPv2 and DDLP. In Table 11 we report the performance on the `Balls-Interaction` dataset by removing the corresponding component out of: using the standard learned positional embeddings in PINT instead of the relational positional encoding and using the standard DLP encoder for all frames instead of tracking. Observe that the full model achieves the best performance, and that tracking the particles is crucial for accurate dynamics modeling We perform further ablation of changing the filtering heuristic of the prior proposals to the distance-heuristic used in the original DLP, instead of filtering based on the SSM covariance.

## J   Failure Cases

Our method is not without limitations – sometimes objects are missed, get deformed or vanish, as we demonstrate in Figures 42 and 43. However, these phenomena are less frequent than in other methods, as

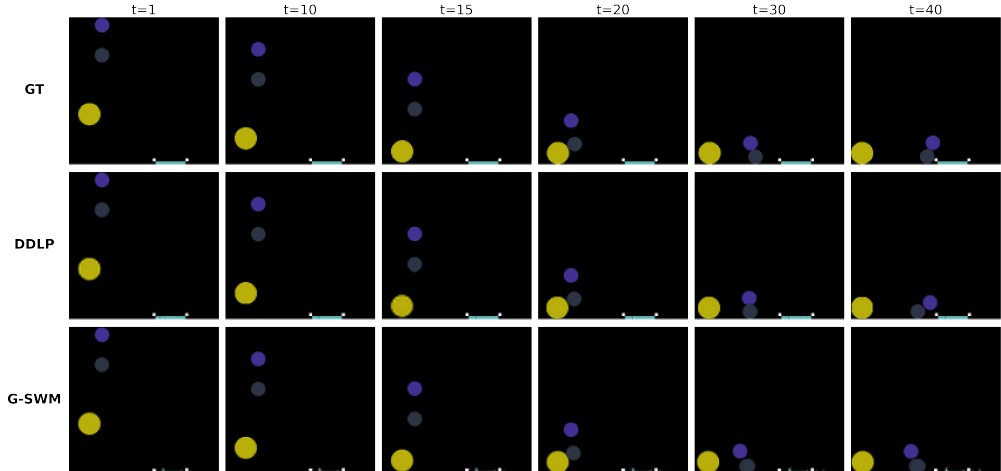

Figure 36: Rollout from the `PHYRE` dataset.

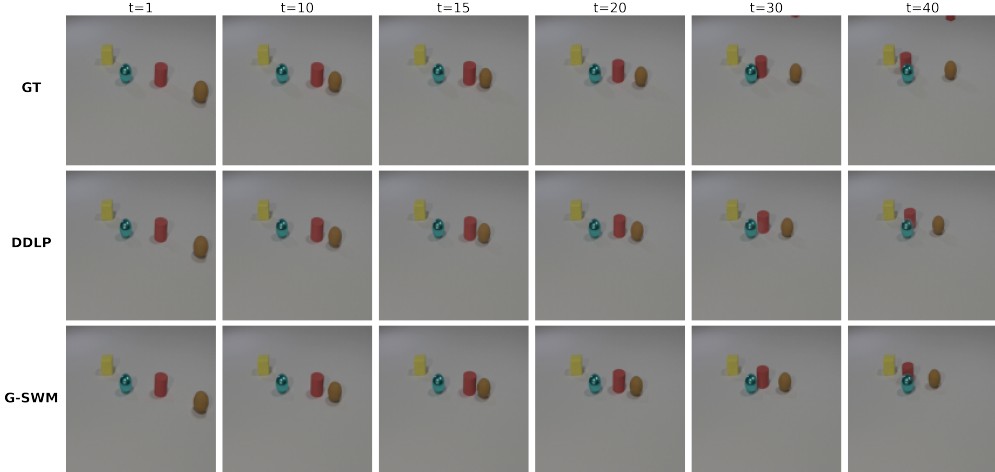

Figure 37: Rollout from the `CLEVRER` dataset.

| | MED10 ↓ | PSNR ↑ | SSIM ↑ | LPIPS ↓ |
|---|---|---|---|---|
| **Full** | **0.149** | **16.74** | **0.789** | **0.189** |
| −No RPE | 0.157 | 16.73 | 0.787 | 0.194 |
| −No Tracking | 2.736 | 11.68 | 0.435 | 0.393 |
| −No NCC | 0.168 | 16.52 | 0.783 | 0.196 |
| −Distance-filtering | 0.163 | 16.43 | 0.786 | 0.191 |

Table 11: Ablation study on the `Balls-Interaction` dataset. In each row, we remove one component from the full model. No RPE - use standard positional embedding instead of relative positional encoding. No Tracking - use the standard single-image DLP encoding for all frames instead of tracking. Distance-filtering - using the distance-heuristic instead of the SSM covariance to filter keypoint proposals. No NCC- - no normalized cross-correlation in the tracking process.

evident by the quantitative results. We believe that scaling-up the model size, e.g., a larger Transformer, may help alleviate these, and perhaps additional tuning of the model hyper-parameters.

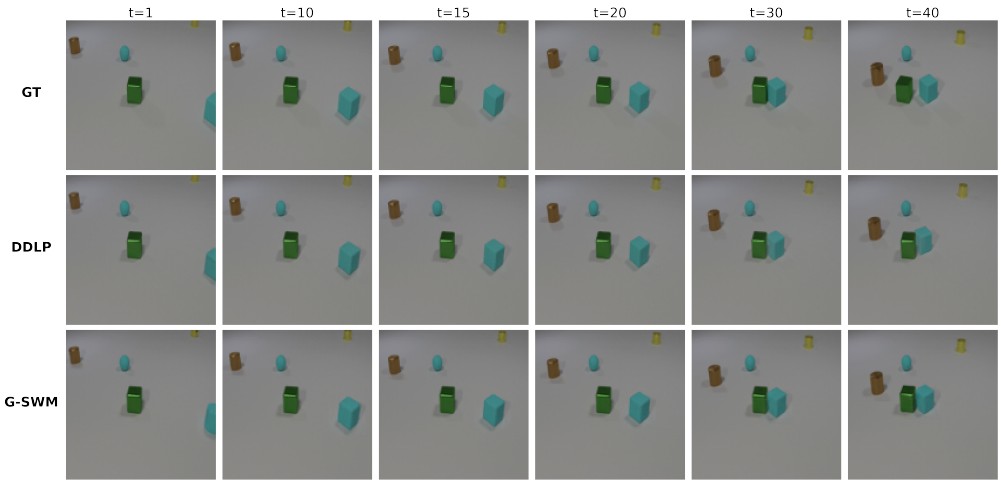

Figure 38: Rollout from the `CLEVRER` dataset.

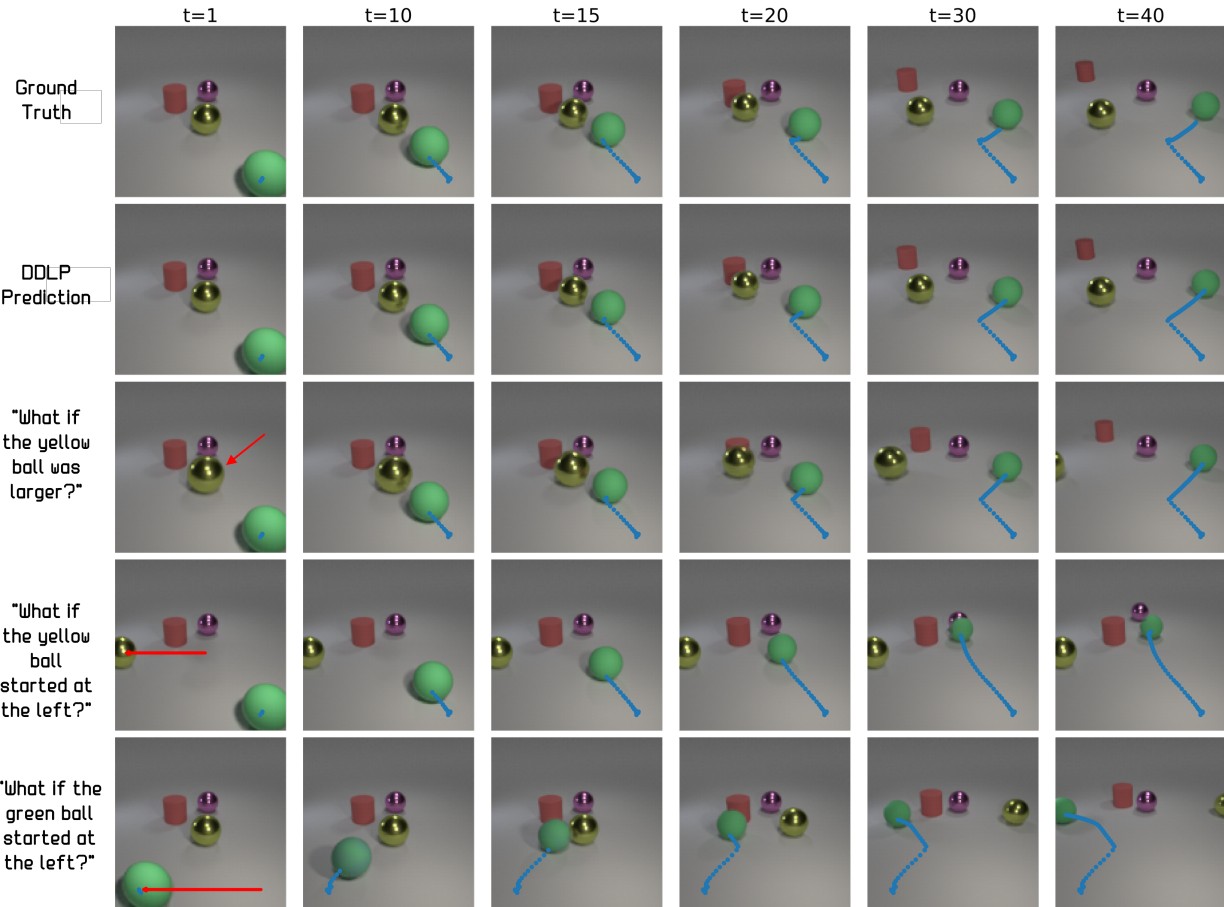

Figure 39: DDLP video prediction on `OBJ3D`. The first row shows the ground-truth frame sequence with the predicted posterior keypoint trajectory (blue). In the second row, we present DDLP's generated particles' trajectory (blue) when conditioned on the first 4 frames, and the decoder's reconstructed sequence from the particles. Row 3-5 present modifications performed in the latent particle space (red arrow) according to "what if...?" questions, and the resulting rollout by the dynamics module. Note the different object dynamics resulted from each modification.

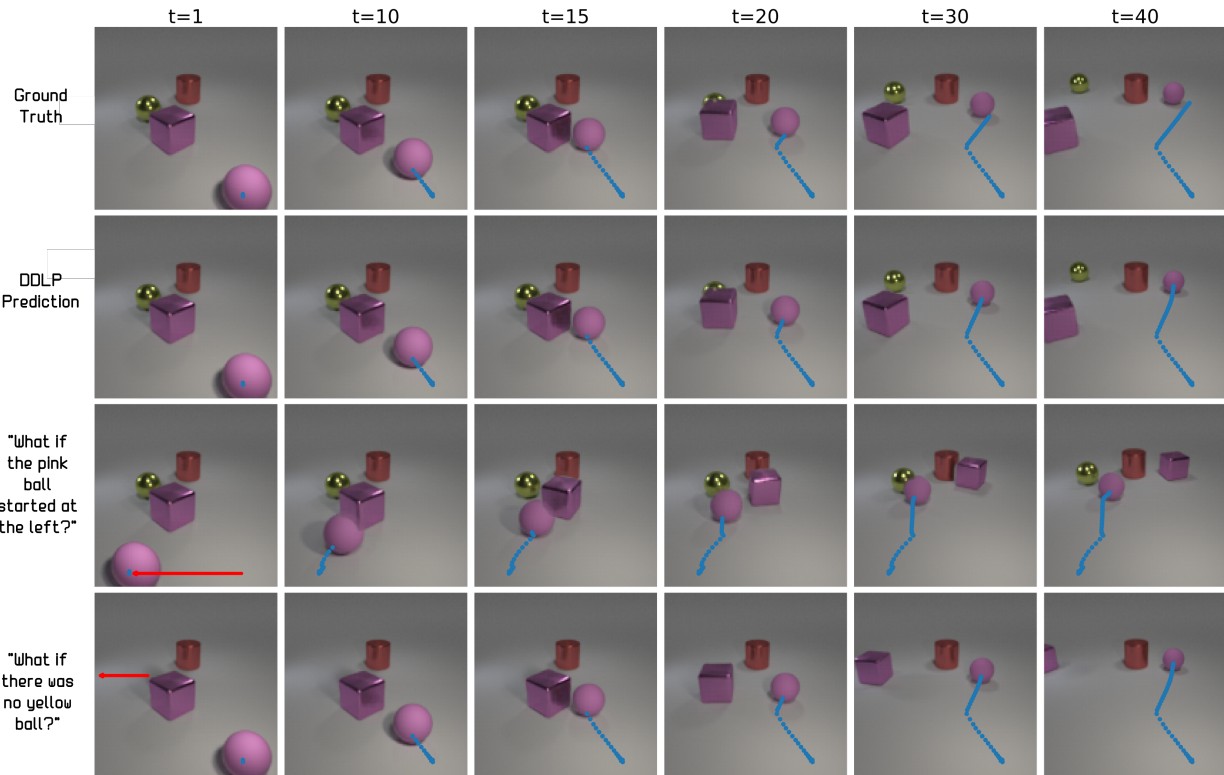

Figure 40: DDLP video prediction on `OBJ3D`. The first row shows the ground-truth frame sequence with the predicted posterior keypoint trajectory (blue). In the second row, we present DDLP's generated particles' trajectory (blue) when conditioned on the first 4 frames, and the decoder's reconstructed sequence from the particles. Row 3-4 present modifications performed in the latent particle space (red arrow) according to "what if...?" questions, and the resulting rollout by the dynamics module. Note the different object dynamics resulted from each modification.

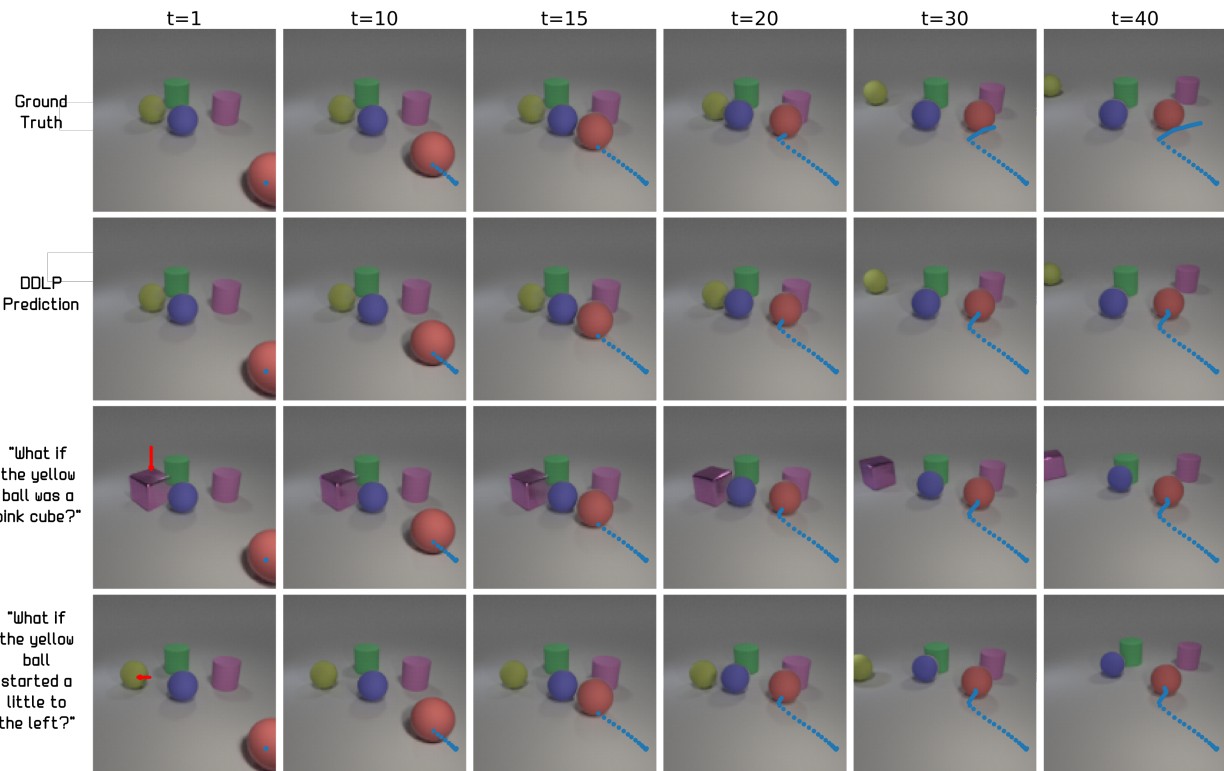

Figure 41: DDLP video prediction on `OBJ3D`. The first row shows the ground-truth frame sequence with the predicted posterior keypoint trajectory (blue). In the second row, we present DDLP's generated particles' trajectory (blue) when conditioned on the first 4 frames, and the decoder's reconstructed sequence from the particles. Row 3-4 present modifications performed in the latent particle space (red arrow) according to "what if...?" questions, and the resulting rollout by the dynamics module. Note the different object dynamics resulted from each modification.

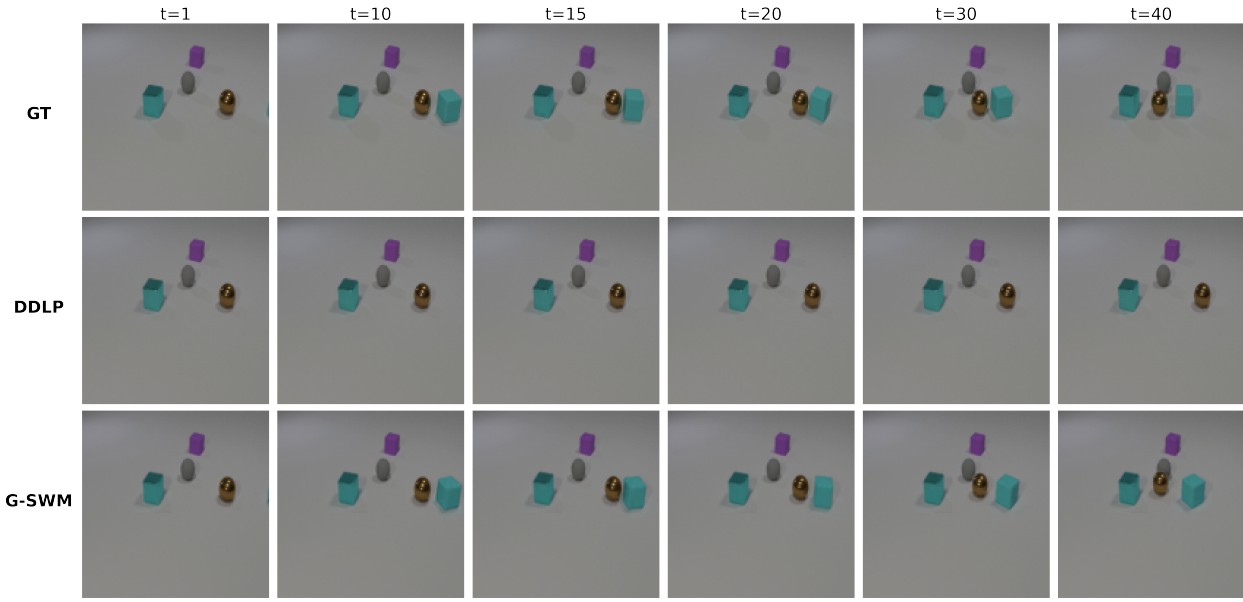

Figure 42: Failure case on CLEVRER. DDLP misses an object, leading to wrong dynamics modeling.

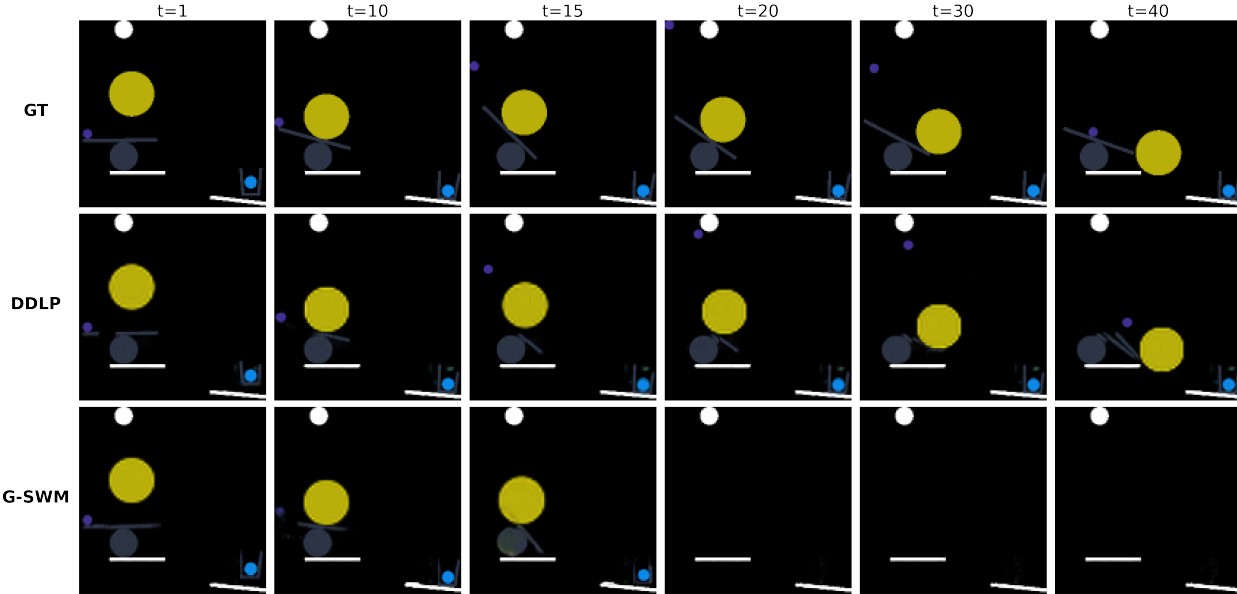

Figure 43: Failure case on PHYRE. DDLP deforms the objects, leading to wrong dynamics modeling.

