# OpenReview forum: "DDLP: Unsupervised Object-centric Video Prediction with Deep Dynamic Latent Particles"
_TMLR — Accepted by TMLR_

### Review · Reviewer_8tEt · 2023-11-09

**Summary Of Contributions:**

This paper extends the Deep Latent Particle (DLP) to Dynamic DLP for future video frame prediction. To this end, the authors introduced several improvements including: a) adding scale, depth, and transparency attributes to the DLP attribute set, b) associating particles between frames using correlation, c) an autoregressive transformer predictor that predicts multiple particles in a single inference, and d) an unconditional particle attributes diffusion process to generate a scene with realistic physical interactions. To demonstrate the effectiveness of the proposed method, the authors conducted experiments on five different datasets, and on tasks of video prediction and unconditional video generation, both of which have achieved superior performance over previous methods.

**Audience:**

Yes

**Broader Impact Concerns:**

The Broader Impact has been sufficiently discussed in the paper.

**Claims And Evidence:**

Yes

**Requested Changes:**

In general, I do not see significant weakness on the method itself except for the question about the temporal-spatial bias fuse.
The authors can focus on improving the organization of the paper, e.g.
- Reprioritize the content in the appendix and the main paper.
- Make the method sections more clear with better illustration, currently the figures are confusing while the text are less well organized, e.g. the Figure3 is confusing to add MLPs right to the other attributes, which can confuse readers about whether the visual features encoding need to take other attributes as input (I assume not).

**Strengths And Weaknesses:**

Strengths:
+ As a extended paper for its conference version, the authors have clearly introduced the improvement brought by this paper and the motivation behind, as well as the perquisite knowledge to understand the topic.
+ The autoregressive transformer in Sec.4.4 is well designed, and spatial-temporal discriminativeness is considered and handled as spatial-temporal positional bias.
+ The efficiency and the performance of the proposed method are relatively better significantly than existing works.
+ The appendix compensate some finer details such as the model structures and the model training details.

Weakness:
- The overall organization of the paper could be improved. Some details are in the appendix instead of the main paper, e.g. the detailed network structure and the detailed training losses. Instead, I would suggest to move some of the content in the main paper to the appendix, to ensure that the reader can fully understand "what" is the model and "how" it works, "why" some components are designed as it is can be optional in appendix. For example, the authors can introduce the added attributes more briefly.
- The spatial-temporal bias design looks inconsistent to it motivation to me. The authors using "+" to fuse the temporal and spatial (order) biases, which is not a one-to-one mapping, i.e. the same added bias may represent to different temporal and order index. Instead of adding, multiplication makes more sense to me since it ensure a one-to-one mapping.
- More realistic data such as the BAIR Robotic Pushing can be added to verify the method in real world.

---

> ### Author Response · Authors · 2023-12-08
> **Thank you for the detailed review - response**
>
> Thank you for your detailed review and effort to improve our paper.
>
> **Organization** - to your suggestion, we will move the requested information to the main text and shorten the attributes section in the main text.
>
> **Attributes figure (Figure 3)** - the reviewer is correct that we do not explicitly use the other attributes to encode the visual features of particles; however, some attribute are implicitly used to extract these features – the position (keypoint) and scale attributes determine the area from which the visual features are encoded.
>
> **Spatio-temporal bias** - thank you for the interesting suggestion. In fact, we tried multiple variations of encoding this bias, including multiplication and even additive variations similar to FiLM [1], however, none has outperformed the simple addition of the spatial and temporal bias. There are several other works that similarly to our approach, combine different types of positional embeddings via addition [2,3].
>
> **BAIR dataset** - The BAIR robotic pushing dataset does not fit our framework as the transition between frames is stochastic (i.e., by taking a random action between frames), making it different from the datasets we used in this work, which are physics-based predictions (i.e., the dynamics are not governed by unobserved external actions). Future work can explore extending DDLP to support stochastic actions.
>
>
> [1] Perez, Ethan, et al. "Film: Visual reasoning with a general conditioning layer." Proceedings of the AAAI conference on artificial intelligence. Vol. 32. No. 1. 2018. - https://arxiv.org/abs/1709.07871
>
> [2] Cao, Jinkun, Hao Wu, and Kris Kitani. "Track targets by dense spatio-temporal position encoding." arXiv preprint arXiv:2210.09455 (2022). - https://arxiv.org/abs/2210.09455
>
> [3] He, Tianyu, et al. "Dense Interaction Learning for Video-based Person Re-identification." 2021 IEEE/CVF International Conference on Computer Vision (ICCV). IEEE, 2021. - https://arxiv.org/abs/2103.09013

---

### Review · Reviewer_YXT5 · 2023-11-19

**Summary Of Contributions:**

In this manuscript, the authors study the problem of Object-centric video prediction. The core contributions of this work are threefold.

First, propose DLPv2, which is an improved version of DLP, enhanced with more explicit attributes of the particles like scale, depth, and transparency. These attributes can be used to control the generation process in a more granular manner.

Second, they re-design the architecture of the transformer, especially the attention matrix with modified position encoding, to make the overall framework more suitable to model the interaction among time and across particles.

Third, they introduce tracking into the framework to align the keypoints across different frames to make the particles more stable, leading to more consistent predictions.

The experiments indeed show the advances of the proposed DDLP, as well as its new ability to generate “what if …” prediction.

**Audience:**

Yes

**Broader Impact Concerns:**

As the core contributions are the design of the new framework, I do not see Boarder Impact Concerns.

**Claims And Evidence:**

Yes

**Requested Changes:**

Some points that are unclear.

1, On page 6, it is said that “(1) the posterior keypoints from the previous step serve as anchors, from which an offset will be learned to the current step as illustrated in Figure 2”. However, in Fig. 2 there is no illustration of the offset.

2, In the "what if" experiments, we can indeed change the latent position code to force the ball to the left-right. But I’m not sure how to deal with the Burn-in frames, as we still need some real images of where the ball is located on the left side (correct me if is there anything I misunderstanding).

3, For the explicit modeling of attributes of the particles, do we need the ground truth attributions? It is unclear to me how to train these attributes to bind their behaviors to the decoder. That is to say: how do we ensure that changing the attributes of the particles can lead to a corresponding change of the generated image?

**Strengths And Weaknesses:**

Strengths:

1, The overall writing of this manuscript is clear and easy to follow.

2, Explicitly modeling the attributes of the particles is a good idea for video prediction, as it provide us more interpretable intermediate representation.

3, The proposed DDLP achieves better at lone-term consistency generation, which aligns well with the intuition behind the design.

4, Extending the latent particles to unconditional generation with diffusion model is interesting and may find more audience in the community.

Weaknesses

1, I would suggest making the main part of this manuscript more self-contained instead of frequently referring to the supplemental materials. For example, on page 7, it is said that “The relative positional bias is illustrated in Figure 22.”. One needs to jump almost to the end of the manuscript to find the illustration and then jump back. It would be better to just have a simple illustration in the main body and then refer the details to the supplemental materials if readers are interested in the details.

2, Some parts of the technological design are unclear. I found it a little hard to fully understand the details. Please refer to the “Requested Changes” part.

---

> ### Author Response · Authors · 2023-12-08
> **Thank you for insightful review - response**
>
> We are grateful for your insightful review and for recognizing the contribution of our work.
> In the following, we address the required clarifications.
>
> **Details vs. figures** - We will follow your suggestion regarding Figure 22, and reorganize this part in the main text.
>
> **Method figure** - in Figure 2 we illustrate the overall architecture. Note that there are arrows from the encoded particles at time-step $t-1$ entering the particle encoder at time-step $t$, indicating the tracking process. Modeling the offset is a design choice that is not necessary, but intuitive and leads to faster convergence. We did not explicitly illustrate the offset modeling to make the illustration cleaner and less cluttered.
>
> **“What-if”** - thank you for raising this important point. In practice, we apply the same modification (e.g., move a particle by Δx=+3) for the first “burn-in frames” (=4 in our work). This is possible since our assumption is that the delta between frames is small, so objects’ displacement over the span of “burn-in frames” is small. We will better clarify this in the revision.
>
> **Attribute modeling** - our method is **completely unsupervised**, there is no need for any ground-truth supervision of attributes, and in fact, most of the datasets used in our paper do not provide such labels. Our approach builds on DLP [1], which utilizes spatial-softmax (SSM) [2] to extract keypoints (the position attribute) in a differentiable manner. In addition, we use neural networks conditioned on the input image or glimpses to encode the other attributes such as scale. To ensure attributes accurately represent their assigned properties (e.g., using the scale attribute to define the bounding box around the particle), we employ a Spatial Transformer Network (STN) [3], which makes the stitching process of the final reconstructed image differentiable. STN takes in the keypoint (position attribute) and the scale attribute, and moves each decoded glimpse to its inferred position (keypoint) on the canvas and rescales it according to the scale attribute by utilizing bilinear sampling. STN’s functions are natively implemented in PyTorch [4]. Finally, we factor the transparency and depth attributes as described (and illustrated in code) in Appendix D (see Figure 17 for code). This process is similar to other patch-based object-centric models, such as G-SWM [5]. In conclusion, all attributes are learned in a differentiable manner with the objective of reconstructing the original image, without any external supervision such as ground-truth attributes. By learning to reconstruct the images, which are essentially object-based, the inductive bias in DLP (explained above) steers the training to learn the correct attributes, in an unsupervised manner.
>
>
> [1] Daniel, Tal, and Aviv Tamar. "Unsupervised Image Representation Learning with Deep Latent Particles." International Conference on Machine Learning. PMLR, 2022. - https://arxiv.org/abs/2205.15821
>
> [2] Jakab, Tomas, et al. "Unsupervised learning of object landmarks through conditional image generation." Advances in neural information processing systems 31 (2018). - https://arxiv.org/abs/1806.07823
>
> [3] Jaderberg, Max, Karen Simonyan, and Andrew Zisserman. "Spatial transformer networks." Advances in neural information processing systems 28 (2015). - https://arxiv.org/abs/1506.02025
>
> [4] https://pytorch.org/tutorials/intermediate/spatial_transformer_tutorial.html
>
> [5] Lin, Zhixuan, et al. "Improving generative imagination in object-centric world models." International Conference on Machine Learning. PMLR, 2020. - https://arxiv.org/abs/2010.02054

---

> > ### Comment · Reviewer_YXT5 · 2024-01-04
> > **Thanks authors for the clarification**
> >
> > As per the authors' response and promise, I no longer have any significant concerns regarding this manuscript.

---

### Review · Reviewer_uzkC · 2023-12-06

**Summary Of Contributions:**

The work proposes a new framework for object-centric video prediction based on Deep Latent Particles (DLP). The contributions include improvement of DLP for single images by including additional attributes such as scale, depth and transparency. It proposes inclusion of particle tracking posterior to efficiently track particles among different frames. Further it also proposes Particle interaction Transformer which enriches the prior distribution of particles with temporal information. Overall, the work improves over prior works based on LPIPS metric score.

**Audience:**

Yes

**Broader Impact Concerns:**

All concerns are already present in the paper.

**Claims And Evidence:**

Yes

**Requested Changes:**

1. Suggested changes in writing:
    a. Introduction lacks motivation of why DDLP is required.
    b. I feel DLPv2 details should be in the main paper and not in appendix as it is also the contribution. (D1-4)
    c. Also the optimisation strategy should be in the main paper: section E, Optimisation objective (page 31). This section helps the reader
        to understand the final optimization loss for reconstruction.
    d. Figure 8, caption: GSWN tends to “deform object” over time, maybe a more elaborative caption would help. I can observe a slightly
         poor reconstruction with red circle, but a proper caption with more details as what is deformed here e.g. poor edges etc would help.
    e. Authors mention SAVi++ is limited to synthetic datasets [page 36], which seems to be incorrect as SAVi++ works with real world
        scenes on Waymo Open dataset.

2. Experimental changes:
   a. Figure 7 shows only decomposition of DLPv2 but no comparison with DLP. Would be nice to see the qualitative comparison with
       DLPv2.

**Strengths And Weaknesses:**

Strength:
1. Lightweight video prediction that can work with many objects
2. Can perform ‘what-if’ generation by playing around with the latent representations which are interpretable (wrt position, shape, depth etc)
3. Can also perform video generation with a diffusion model learned on the latent particles.

Weakness:
1. Ablation experiments -
   a. DLPv2 is a major contribution for improved performance, however we do not see the comparison of method with DLPv2 in main
       results in table 2. It should work as a strong baseline and should be present in the main table.
   b. Ablation with important components in a standard way. I feel the 3 major components are DLPv2, particle tracking in posterior
       (encoder), particle interaction transformer as presented in the paper. The ablations should have results with these three explicitly. I
       see in table 2 ablation there is -DLP (No Tracking) which I assume should be DLPv2 ? Apart from that other components ablation is
       really important for the main paper.

2. Main table results - should also include PSNR, SSIM scores for proper evaluation. Although these numbers are in appendix, it also makes sense to put them in the main table.

3. Comparison with SlotFormer on CLEVRER, PHYRE and Balls-Interaction dataset. Also I may misunderstand something here, but from the published slotformer work, the LPIPS score for obj3d dataset is 0.080 while the reported number in the manuscript is 0.13+-0.05. They also present numbers for CLEVRER dataset and achieve 0.11 LPIPS score, and the proposed work DDLP achieves 0.13+-8.

4. To show error in prediction, authors presented MED10 metric which computes mean euclidean distance error of predicted balls positions for first 10 steps of generation. Would also be nice to see scores and comparisons for more steps.

5. Discussion on why object centric representation based method is needed. There are a range of work for video prediction without object centric representation learning e.g. RIVER[1], VDM[2] etc. Authors should also motive the need for object centric representation learning for video prediction.

6. Ablation with positional embeddings -
The choice of choosing positional bias as $b_i + b_j$, why not $b_i \times b_j$, or some other formulation ? Comparison with just using $b_i$ or just using $b_j$, to understand importance of spatial and temporal positional embeddings separately.

[1] Efficient Video Prediction via Sparsely Conditioned Flow Matching, ICCV 23
[2] Video diffusion models. arXiv preprint arXiv:2204.03458, 2022.

---

> ### Author Response · Authors · 2023-12-08
> **Thank you for your time - response**
>
> Thank you for the time invested in reviewing our paper, we appreciate it. In the following, we address the reviewer comments.
>
> **Ablation (DLPv2 as baseline)** - we do not understand the reviewer’s comment on DLPv2 as baseline – DLPv2 is a single-image object-centric decomposition method on which DDLP builds upon by adding the several components discussed in the paper, such as the tracking and dynamic modules. The ablation “DLP (No Tracking)” in Table 2 (right) is plain DLPv2 without tracking (which doesn’t work, as DLPv2 only provides single-image decomposition to unordered particles). The tracking procedure is tailored for DLPv2, thus it is not possible to perform an ablation of tracking without DLPv2.
>
> **Ablation (3 components)** - first, we establish DLPv2 superiority over DLP in the single-image setting, both quantitatively (Table 7) and qualitatively (Figures 25-26), in Appendix I.1.1 (pages 36-37). In addition, Table 2 (right) presents ablations of removing the tracking module (“DLP (No Tracking)”, which is plain DLPv2 with a dynamics module) and removing the spatio-temporal bias from the dynamics module. We do not understand how it is possible to model video dynamics without a dynamics module, thus, there is no ablation of removing the dynamics module. To evaluate our method quantitatively we therefore compare it with SOTA object-centric methods such as G-SWM and SlotFormer.
>
> **Ablation (positional embeddings)** - we tried multiple variations of encoding this bias, including multiplication and even additive variations similar to FiLM [5], however, none has outperformed the simple addition of the spatial and temporal bias. There are several other works that similarly to our approach, combine different types of positional embeddings via addition [6,7].
>
> **PSNR and SSIM in Table 2**- we will move the metrics to the main table.  We reported LPIPS results in the main text only due to space considerations. We chose to focus on LPIPS as it is the most important result, as we shall now explain. LPIPS is a perceptual measure, while SSIM and PSNR are distortion measures. It is well known that LPIPS is more aligned with human perception and better accounts for small variations such as object deformation and blurriness [1,2].
>
> **Comparison with SlotFormer** - we provide an extended comparison with SlotFormer [1] in Appendix H (pages 34-36). *OBJ3D* -  Please note that in the original SlotFormer paper, their model is trained on 64x64 resolution, while we train our model on a higher resolution of 128x128, for which we report the results in Table 2 and Table 8. We also trained our model on a 64x64 and compared with their pre-trained model, and the results are reported in Table 6. It can be seen that DDLP outperforms SlotFormer both on 128x128 and 64x64 resolutions. It is important to note that the results reported in [1] are on the validation set – the same dataset that was used for tuning hyperparameters. To ensure consistency with our paper, we used their pre-trained model to evaluate the reported metrics on the validation set and also computed the same metrics on the test set (which was not used for training nor hyperparameter tuning), as discussed in Appendix H (page 35, first pargaraph). Our paper only reports results for the test sets. Furthermore, in Appendix H we also explain why it is unfair to directly compare DDLP with SlotFormer on PHYRE and CLEVRER. In short, they perform different pre-processing steps that are required for SlotFormer to work on these datasets, steps which are not required for DDLP (or G-SWM) to work.
>
> **MED10** - we follow the standard evaluation protocol for the Balls-Interaction dataset established in previous works [3, 4], and provide LPIPS, PSNR and SSIM scores for longer horizons in Table 2 and Table 9.
>
> **Motivation (why are object-centric representations required?)** - our work falls within the well-established object-centric literature, and we compare our method to the SOTA methods in this field. Thank you for the interesting references. Previous studies [1, 3, 4] have shown the benefits of the object-centric inductive bias over non-object-centric methods, outperforming them on datasets with multiple objects, and thus our comparison is focused on object-centric baselines and datasets.

---

> > ### Author Response · Authors · 2023-12-08
> > **continued response**
> >
> > **Motivation (why DDLP?)** - In the introduction, we emphasize the advantages of DDLP over other object-centric models, and rigorously demonstrate them throughout the paper. In general, DDLP strikes a balance between a lightweight representation for all patches and a
> > heavy representation for a small number of slots, while also boasting a lower training memory requirement. In addition, the DLP representation is interpretable and enables to control the generation by modifying latent space properties of particles. Finally, we demonstrate two novel applications made possible thanks to DDLP - “What if…?” video prediction and unconditional object-centric video generation with diffusion. Both of these novel applications build on unique properties of DDLP, and we do not know how they can be executed with previous methods.
> >
> > **DLP vs. DLPv2** -  we compare DLP and DLPv2 in the single-image setting, both quantitatively (Table 7) and qualitatively (Figures 25-26), in Appendix I.1.1 (pages 36-37).
> >
> > **Presentation** - we will provide more details on DLPv2 and the optimization objective of DDLP in the main text. We will extend Figure 8’s caption with more details.
> >
> > **SAVI++** - thank you for pointing this out, we did not mean that SAVI++ can not work on real-world datasets, but that its performance is limited in these cases, as thoroughly investigated in [8]. In our personal experience, we also faced difficulties getting SAVI++ to perform well on the Traffic dataset. We will relax this claim in the text.
> >
> >
> > [1] Wu, Ziyi, et al. "SlotFormer: Unsupervised Visual Dynamics Simulation with Object-Centric Models." ICLR 2022 - https://arxiv.org/abs/2210.05861
> >
> > [2] Blau, Yochai, and Tomer Michaeli. "The perception-distortion tradeoff." CVPR. 2018 - https://arxiv.org/abs/1711.06077
> >
> > [3] Lin, Zhixuan, et al. "Improving generative imagination in object-centric world models." International Conference on Machine Learning. PMLR, 2020. - https://arxiv.org/abs/2010.02054
> >
> > [4] Jiang, Jindong, et al. "SCALOR: Generative World Models with Scalable Object Representations." ICLR 2019 - https://arxiv.org/abs/1910.02384
> >
> > [5] Perez, Ethan, et al. "Film: Visual reasoning with a general conditioning layer." Proceedings of the AAAI conference on artificial intelligence. Vol. 32. No. 1. 2018. - https://arxiv.org/abs/1709.07871
> >
> > [6] Cao, Jinkun, Hao Wu, and Kris Kitani. "Track targets by dense spatio-temporal position encoding." arXiv preprint arXiv:2210.09455 (2022). - https://arxiv.org/abs/2210.09455
> >
> > [7] He, Tianyu, et al. "Dense Interaction Learning for Video-based Person Re-identification." 2021 IEEE/CVF International Conference on Computer Vision (ICCV). IEEE, 2021. - https://arxiv.org/abs/2103.09013
> >
> > [8] Seitzer, Maximilian et al. “Bridging the gap to real-world object-centric learning.” ICLR, 2023. - https://openreview.net/forum?id=b9tUk-f_aG

---

> ### Comment · Reviewer_uzkC · 2023-12-08
> **Thanks for the response**
>
> DLPv2 baseline confusion: Indeed DLPv2 cannot work out of the box for next frame prediction. However a naive DLPv2 baseline can be obtained by replacing the transformer dynamics module with say a feed forward network to predict next priors, and without any tracking at the encoder particles. Such naive baseline can be used as comparison, which can show importance of the transformer module that better predicts the next priors.

---

> > ### Author Response · Authors · 2023-12-09
> > **Thank you for the prompt response!**
> >
> > Thank you for the prompt response and for the clarification regarding the baseline. The original DLP paper [1] utilized a simple GNN over the unordered latent particles, and as discussed in [1] (Section 6, page 8, “Limitations and Future Work''), the performance of the GNN is limited to datasets with simple dynamics such as the Traffic dataset, but does not work on datasets with complex dynamics such as CLEVRER. Previous works have also reached similar conclusions and utilize stronger architectures with inductive biases that consider interactions. For example, G-SWM [2] combines an interaction network with an RNN, and SlotFormer [3] utilizes a Transformer with temporal embeddings. In light of the strong performance over G-SWM and SlotFormer, we are not sure what purpose replacing the Transformer with an MLP serves, but we are willing to set-up such an experiment if the reviewer finds it beneficial. We would like to emphasize that we are not claiming novelty over using a Transformer as a dynamics module (which works well in other methods), but that our approach (DLPv2 + tracking + modified attention + spatio-temporal embeddings) enables the integration of the Transformer with latent particles.
> >
> >
> > [1] Daniel, Tal, and Aviv Tamar. "Unsupervised Image Representation Learning with Deep Latent Particles." International Conference on Machine Learning. PMLR, 2022. - https://arxiv.org/abs/2205.15821
> >
> > [2] Lin, Zhixuan, et al. "Improving generative imagination in object-centric world models." International Conference on Machine Learning. PMLR, 2020. - https://arxiv.org/abs/2010.02054
> >
> > [3] Wu, Ziyi, et al. "SlotFormer: Unsupervised Visual Dynamics Simulation with Object-Centric Models." ICLR 2022 - https://arxiv.org/abs/2210.05861

---

### Decision · Action_Editor_HJ8V · 2024-01-23

**Recommendation:** Accept with minor revision

**Comment:**

The paper was reviewed by three reviewers. After the rebuttal period, all three reviewers recommended "leaning accept''.

One remaining concern is that the main text should be made more self-contained by moving some contents from the supplementary materials to the main text.

The reviewers were positive about the proposed method in general, including the design of the model architecture, the what-if generation, and the diffusion-based generation. There were initially some concerns about experiments, including ablation studies and additional results, but these concerns were all successfully addressed in the rebuttal period.

Overall, I think this paper makes a useful contribution to video prediction, and I recommend accept with minor revision.

**Audience:**

The paper is of interest to researchers working on video prediction and generation.

**Claims And Evidence:**

This paper proposes an object-centric video prediction method by extending the deep latent particle (DLP) model. The proposed method, deep dynamic latent particles model, achieves SOTA results on several datasets. The model allows what-if generation and diffusion-based generation.

The claims made in the submission are supported by experimental evidences.